# A novel mouse model of mitochondrial disease exhibits juvenile-onset severe neurological impairment due to parvalbumin cell mitochondrial dysfunction

Elizaveta A. Olkhova [1,2], Carla Bradshaw[1], Alasdair Blain[1,2], Debora Alvim[1,2], Doug M. Turnbull[1,2,3,4], Fiona E. N. LeBeau [5], Yi Shiau Ng[1,2,3,4], Gráinne S. Gorman [1,2,3,4✉] & Nichola Z. Lax[1,2]

Mitochondrial diseases comprise a common group of neurometabolic disorders resulting from OXPHOS defects, that may manifest with neurological impairments, for which there are currently no disease-modifying therapies. Previous studies suggest inhibitory interneuron susceptibility to mitochondrial impairment, especially of parvalbumin-expressing interneurons (PV+). We have developed a mouse model of mitochondrial dysfunction specifically in PV+ cells via conditional *Tfam* knockout, that exhibited a juvenile-onset progressive phenotype characterised by cognitive deficits, anxiety-like behaviour, head-nodding, stargazing, ataxia, and reduced lifespan. A brain region-dependent decrease of OXPHOS complexes I and IV in PV+ neurons was detected, with Purkinje neurons being most affected. We validated these findings in a neuropathological study of patients with pathogenic mtDNA and *POLG* variants showing PV+ interneuron loss and deficiencies in complexes I and IV. This mouse model offers a drug screening platform to propel the discovery of therapeutics to treat severe neurological impairment due to mitochondrial dysfunction.

[1] Wellcome Centre for Mitochondrial Research, Faculty of Medical Sciences, Newcastle University, Framlington Place, Newcastle upon Tyne NE2 4HH, UK. [2] Translational and Clinical Research Institute, Faculty of Medical Sciences, Newcastle University, Framlington Place, Newcastle upon Tyne NE2 4HH, UK. [3] NHS Highly Specialised Service for Rare Mitochondrial Disorders, Newcastle upon Tyne Hospitals NHS Foundation Trust, Newcastle upon Tyne NE2 4HH, UK. [4] NIHR Newcastle Biomedical Research Centre, Biomedical Research Building, Campus for Ageing and Vitality, Newcastle upon Tyne NE4 5PL, UK. [5] Biosciences Institute, Faculty of Medical Sciences, Newcastle University, Framlington Place, Newcastle upon Tyne NE2 4HH, UK. ✉email: grainne.gorman@newcastle.ac.uk

Mitochondrial diseases are the most common inherited neurometabolic disorders characterised by oxidative phosphorylation (OXPHOS) impairments due to either pathogenic variants in mitochondrial DNA (mtDNA) or nuclear-encoded DNA[1]. The central nervous system is frequently affected in patients with mitochondrial disease, commonly manifesting with cerebellar ataxia, stroke-like episodes, epilepsy, extrapyramidal movement disorders, and cognitive impairment[1,2]. Currently, there are no cures for mitochondrial disease and treatment strategies are limited primarily to ameliorating symptoms.

To develop effective treatments, it is crucial to delineate the mechanisms contributing to debilitating symptoms such as ataxia and epilepsy in mitochondrial disease. Previous neuropathological studies confirm the emergence of focal necrotic lesions affecting the neocortex and cerebellum that are associated with ictal activity on electroencephalogram and T2-weighted hyperintensities on cranial magnetic resonance imaging[2–6]. Apart from stroke-like lesions occurring in the cerebral cortex, lesions have additionally been reported in the cerebellum of patients with mitochondrial encephalomyopathy lactic-acidosis and stroke-like episodes (MELAS) due to mtDNA m.3243 A > G and biallelic *POLG* pathogenic variants[6–8]. *Post-mortem* neuropathological findings reveal loss of large inhibitory Purkinje neurons of the cerebellum, as well as the dentate nucleus neurons, concomitant with surviving neurons exhibiting extensive complex I deficiencies, that are thought to underlie cerebellar ataxia[9–11].

The reported loss of GABAergic inhibitory neurons, combined with mitochondrial dysfunction in the surviving inhibitory neurons, is hypothesised to underlie changes in the neural circuitry which may promote hyperexcitability in mitochondrial disease[12,13]. OXPHOS deficiencies in surviving pyramidal neurons exhibit lower levels of severity in comparison to the inhibitory interneurons and Purkinje neurons, further implicating inhibitory neuron predilection to mitochondrial dysfunction[13].

Interrogating the susceptibility of the specific subclasses of inhibitory neurons to neurodegeneration and severe mitochondrial dysfunction is critical to delineate the pathophysiology of neurological dysfunction arising in mitochondrial diseases. Because the main and most common subtype of inhibitory neurons, termed parvalbumin-expressing (PV[+]) cells, possess unique biophysical properties, it may render them highly susceptible to bioenergy crisis[14], we hypothesised that PV[+] cell dysfunction may play a key role in mitochondrial disease. Recently, our group has demonstrated that PV[+] interneurons are indeed subject to selective loss and extensive OXPHOS deficiencies in contrast to calretinin-expressing interneurons in Alpers' syndrome[15]. The majority of PV[+] cells are fast-spiking cells and provide perisomatic inhibition to the excitatory cells and thus regulate neuronal network activity[16]. In rodents, fast-spiking interneurons contain a higher concentration of cytochrome *c* oxidase in comparison to excitatory pyramidal neurons[17]. More recent evidence suggests that mouse PV[+] cells may exhibit higher complex IV expression in comparison to somatostatin or calretinin-expressing inhibitory interneurons[18]. Inadequate supply of adenosine triphosphate (ATP) and dysregulation of $Ca^{2+}$ dynamics may be detrimental to PV[+] firing and gamma frequency (20–80 Hz) oscillations[19,20], potentially disinhibiting the neuronal network and leading to excitotoxicity. A mouse model with a conditional knockout of the complex IV subunit in PV[+] cells has demonstrated that mitochondrial dysfunction was sufficient to alter the firing properties of PV[+] cells and cause an imbalance of excitation/inhibition in the cortex and hippocampus of mice, resulting in sociability and sensory gating impairments[18].

The *Tfam* gene encodes for a mitochondrial transcription factor A, necessary for transcription initiation and replication of mtDNA, and its loss results in mtDNA depletion[21,22]. To delineate the phenotypic and neuropathological effects of mitochondrial dysfunction in PV[+] cells in vivo, we generated a mouse model which harboured a conditional knockout of nuclear-encoded *Tfam* specifically in PV[+] neurons via *cre-loxP* system. Our mouse model demonstrated a juvenile-onset progressive neurological impairment resembling severe mtDNA and POLG disease, that ultimately led to severe ataxia and premature death at three months of age. Significantly, OXPHOS deficiencies that resulted from mtDNA depletion in PV[+] neurons displayed a strict brain region-dependent hierarchy. Our findings suggest that severe combined OXPHOS deficiencies in Purkinje neurons of the cerebellum compromise neuronal viability, leading to secondary microglial and astrocytic reactivity. The consequences of selective mitochondrial dysfunction in PV[+] inhibitory neurons, although only accounting for a small proportion of all neurons in the brain, are sufficient to cause detrimental juvenile-onset neurological disease, characterised by ataxia and seizures.

## Results

**Conditional *Tfam* knockout in PV-expressing cells causes juvenile-onset of motor deficits and neurological impairment.** We mechanistically explored the hypothesis of whether selective PV[+] cell mitochondrial dysfunction would be sufficient to induce a neurological phenotype reminiscent of human mitochondrial disease[14] by creating a suitable in vivo mitochondrial disease model based on available neuropathological data[15]. This model would allow us to study the events and processes downstream of mtDNA depletion in PV[+] cells. A mouse model was generated using a conditional knockout of the mtDNA maintenance gene *Tfam* specifically in PV[+] cells via the *cre-loxP* recombination system. PV first becomes detectable in the rodent central nervous system postnatally at 10–14 days of age[23–25], which differs between brain regions, reaching its maximal expression at P35[23] in the cortex and P25-27 in the cerebellum[24]. Gradual changes in PV expression render the expression of cre recombinase and subsequent excision of the *Tfam* gene exons 6 and 7 time-dependent, as previously reported in other models using *cre* under the control of *PV* promoter[18].

Daily monitoring and scoring (as described in Supplementary Table 1) determined that symptoms emerged at 8 weeks of age (juvenile-onset) in 90% of all $PV^{cre/+}Tfam^{-/-}$ knockout mice examined. These manifested as motor deficits, including tremors and twitching in both sexes. Heterozygous mice $PV^{cre}Tfam^{+/loxP}$ were indistinguishable from $PV^{cre}Tfam^{+/+}$ littermates throughout life, hence mice with both genotypes were used as littermate controls. Coinciding with the first motor symptoms, juvenile-onset cognitive deficits (Fig. 1a) and anxiety-like behaviour (Fig. 1b) were detected. No statistically significant differences in visual depth perception were observed at 8–9 weeks of age (Fig. 1c), likely due to a large variability in the knockout group rendering this test underpowered. Specifically, knockout mice showed a significantly reduced discrimination index between the novel and familiar objects in the novel object recognition (NOR) test (Fig. 1a), indicative of cognitive impairment. Knockout mice also entered open arms of the elevated plus maze (EPM) less frequently than closed arms in comparison to their littermate controls, demonstrating an anxiety-like phenotype (Fig. 1b).

Overall locomotor activity of the knockout mice was normal and did not differ from the locomotion of littermates at 6 and 8 weeks of age, as assessed by open-field testing (Supplementary Fig. 1a, b), however, mice exhibited hyperlocomotion by 10 weeks of age in the open-field test (Fig. 1d). Mean total distance travelled was significantly greater in the knockout group ($P = 0.0027$, *t*-test), as well as mean total number of foot touches detected ($P = 0.0023$, *t*-test; Supplementary Fig. 1c). Concomitant

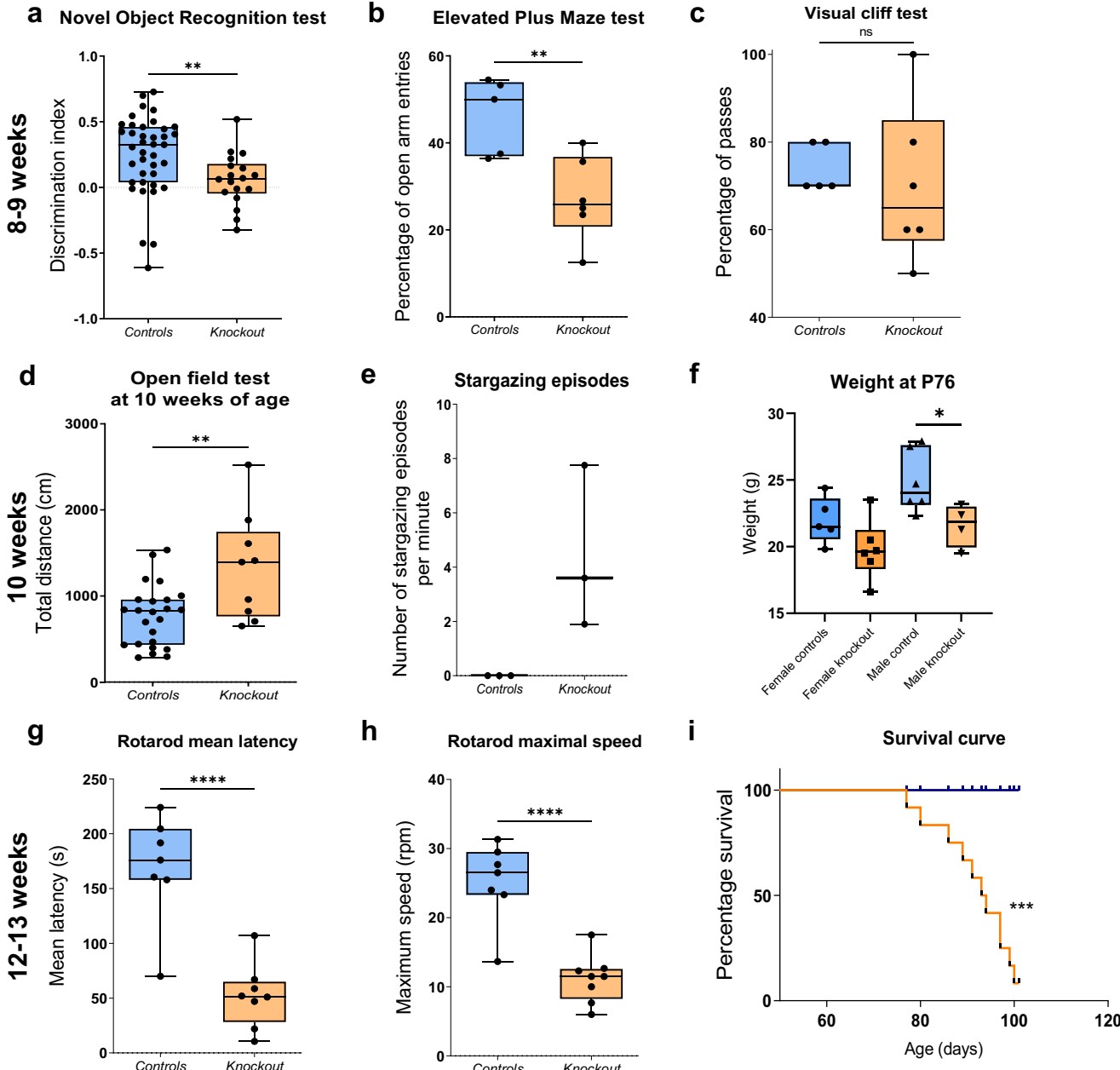

**Fig. 1 $PV^{cre}Tfam^{-/-}$ mice display juvenile-onset progressive symptoms reminiscent of mitochondrial disease at 8, 10 and 12 weeks of age. a** NOR test results demonstrate a four-fold reduction in median discrimination index, measured as the difference in time spent exploring novel and familiar objects, divided by the time spent exploring both objects during the retrieval trial of the test, in the knockout animals ($n = 19$) compared to littermate controls ($n = 38$) ($P = 0.0043$, Mann–Whitney test) at 8 weeks of age. **b** The mean percentage of open arm entries of the EPM is significantly lower in the knockout group (27%) in comparison to the littermate control (46%) group ($P = 0.0079$, $t$-test; $n = 5$ control and $n = 6$ knockout mice) at 8–9 weeks of age. **c** Visual Cliff test shows no visual depth perception impairment in the knockout animals at 8–9 weeks of age in comparison to controls ($P = 0.4493$, Mann–Whitney test; $n = 5$ control and $n = 6$ knockout mice). **d** Open-field test results with distance travelled in littermate controls ($n = 24$) and knockout mice ($n = 9$) at 10 weeks of age. **e** The mean number of stargazing episodes per minute, with two investigators independently quantifying the total number of episodes in a 10-min recording interval at 10 weeks of age. **f** The mean body weight in male knockout mice ($n = 4$) was significantly lower than in male control mice ($n = 6$) and did not reach significance between female mice ($n = 5$ control and $n = 6$ knockout mice) ($P = 0.0413$ and 0.1113, $t$-test, respectively). **g** The mean latency as well as **h** the speed of the rotating rod at which the animal lost its balance was significantly reduced by more than three- and two-fold, respectively, in the knockout mice ($n = 8$), compared to the littermate controls ($n = 7$) ($P < 0.0001$, $t$-test). **i** Kaplan–Meier curve indicates a significant reduction in the survival of the knockout animals ($n = 12$) with a median survival of 94 days in comparison to their littermate controls ($n = 4$) lifespan ($P = 0.0008$, Mantel-Cox log-rank test).

with the hyperactivity in the open-field, continuous head-nodding (modified Racine scale score of 2; Supplementary Table 1) and aberrant stargazing behaviour (Fig. 1e) appeared at 10 weeks of age, similar to those documented in a mouse model of absence-like seizures[26]. The mean weight of male knockout mice was significantly lower than that of the littermate controls and did not reach significance in the female mice group (Fig. 1f; $P = 0.0413$ and 0.1113, $t$-test, respectively).

By 12–13 weeks of age, gait rapidly deteriorated in knockout mice with the observed widening of the hindlimb position for additional balance, moving by writhing on their abdomen and impairment in regaining balance when placed on their backs. Knockout mice had a significantly decreased latency to fall, and a decreased maximal speed on the accelerating rotating rod (rotarod), at which mice lost their balance (Fig. 1g, h). Severe ataxia and reduced ability to maintain adequate body weight in the knockout animals eventually led to a severely reduced lifespan and mice had to be humanely culled at 12–13 weeks of age (Fig. 1i).

**Brain region-dependent OXPHOS subunit expression deficiencies in PV$^+$ neurons in knockout mice.** We selected the following brain regions with high densities of PV$^+$ neurons to be investigated: Purkinje cells and interneurons of the molecular layer of the cerebellum, as well as deep cerebellar nuclei neurons, were analysed due to cerebellar ataxia in the knockout mice; thalamic reticular nucleus (TRN) PV$^+$ neurons due to the star-gazing episodes associated with abnormalities in this region and its roles in gating of signals from thalamus to cortex;[26,27] somatosensory and visual cortical areas, and hippocampus due to its roles in memory processing and retrieval.

Firstly, we checked the voltage-dependent anion channel (VDAC1) porin expression levels between littermate control and knockout mice to investigate changes in total mitochondrial mass. We found porin to be unaltered in the knockout PV$^+$ Purkinje cells, TRN neurons, and cortical and hippocampal PV$^+$ interneurons (Supplementary Fig. 2).

Quantification of OXPHOS subunit expression, including the subunits of complexes I (using an antibody against NADH dehydrogenase [ubiquinone] 1 alpha subcomplex subunit 13, or NDUFA13) and IV (using an antibody against cytochrome $c$ oxidase subunit 4, or COXIV) relative to nuclear-encoded complex II (using an antibody against succinate dehydrogenase A, or SDHA) expression in PV$^+$ neurons (Fig. 2a) revealed intact expression of all subunits examined within the PV$^+$ neurons of littermate control mice. However, there was a brain region-dependent decrease in NDUFA13 and COXIV normalised to SDHA within PV$^+$ neurons in the knockout mice, indicative of OXPHOS deficiencies (Supplementary Fig. 3a–f). No prominent changes in complex II expression were detected across the brain regions investigated (Supplementary Fig. 4).

Purkinje cells of the knockout mice (Fig. 2a) showed the greatest percentage of neurons with combined overall reduction ($z$-score of $\leq -2$) in complexes I (97%) and IV (95%) expression (Fig. 2b). A similar percentage of TRN PV$^+$ neurons showed an overall decrease in complex I expression (94%), however, the proportion of cells with reduced complex IV expression (55%) was smaller than that of Purkinje cells.

In the molecular layer (ML) of the cerebellum, 77% and 38% of PV$^+$ interneurons showed an overall decrease in complexes I and IV, respectively. PV$^+$ interneurons with decreased complex I (27%) and complex IV (20%) expression were detected in the somatosensory and visual cortex. Lastly, in the hippocampal formation 24% of PV$^+$ interneurons exhibited reduced complex I, however, complex IV expression appeared to be almost entirely preserved. On a group level, complex I and IV subunit expression differences did not reach significance in the hippocampus (Supplementary Fig. 3f).

Since Purkinje neurons demonstrated the most severe combined OXPHOS deficiency, we investigated the deep cerebellar nuclei (DCN), which are innervated by Purkinje cells to reduce their excitability. Immunohistochemical staining with anti-glutamate decarboxylase (GAD) 1–2 and anti-PV antibodies revealed that the majority of DCN neuronal cell bodies are devoid of the GABA-synthesising enzyme GAD1-2, except for smaller GAD-expressing putative inhibitory neurons, and show reduced PV expression in comparison to the neighbouring Purkinje neurons (Supplementary Fig. 5). Within the population of DCN neurons, 54% exhibited reduced or deficient complex I expression, while only 12% showed a lower complex IV expression (Fig. 2b). Interestingly, 9% and 24% of the DCN neurons of knockout mice exhibited a concomitant increase in complex I and IV subunit expression, which was not observed in other brain regions to the same extent (Fig. 2b). On a group level, complex I and IV subunit expression differences did not reach significance in the DCN (Supplementary Fig. 3f).

To assess the expression levels of complexes III and mtDNA-encoded subunit of complex IV, antibodies against ubiquinol-cytochrome $c$ reductase core protein 2 (UqCRC2) and cytochrome $c$ oxidase subunit 1 (COXI) were employed, respectively, and their immunofluorescent signal was normalised to porin (Fig. 2c). Interestingly, UqCRC2 demonstrated reduced or deficient levels in only 21% of Purkinje neurons, 5% of TRN, 28% of cortical and 19% of hippocampal PV$^+$ neurons (Fig. 2d). These findings are in accordance with literature suggesting core proteins of complex III such as UqCRC2 remain present in significant amounts in mtDNA-depleted Rho$^0$ cell line[28]. MtDNA-encoded COXI expression patterns (Fig. 2d) revealed similar levels of deficiencies across the brain regions investigated when compared to nuclear DNA-encoded COXIV subunit (Fig. 2b) as outlined above.

Complex V expression was assessed by measuring ATP synthase F$_0$ peripheral stalk subunit B (ATP-B) normalised to porin (Fig. 2e). It was established that ATP-B/porin expression was unaltered in the majority of PV$^+$ neurons of the knockout group in any of the brain regions studied (Fig. 2f), analogous to literature reports of intact assembly of this subunit in cell lines devoid of mtDNA such as Rho$^0$ cells[28,29]. Purkinje neurons showed reduced or deficient ATP-B expression in 9% of cells, whereas 7% of neurons showed overexpression of ATP-B (Fig. 2f).

**mtDNA depletion in Purkinje cells.** TFAM binds and coats mtDNA in a high molar ratio and packages it into nucleoids, participates in transcription initiation, and is essential in the maintenance of mtDNA copy number[22]. TFAM loss from specific tissues or cells has been shown to result in mtDNA depletion[30]. We therefore isolated PV$^+$ Purkinje cells from frozen cerebellar sections to measure mtDNA copy number. MtDNA copy number was significantly decreased in Purkinje cells isolated from knockout mice aged 12–13 weeks in comparison to the littermate control group by approximately 85% (Supplementary Fig. 6).

**Complex I deficiency is milder in Purkinje neuron axonal terminals in deep cerebellar nuclei in comparison to severe complex I deficiencies in Purkinje cell bodies.** Previously published data identified that OXPHOS complex I deficiency was greater in the soma of Purkinje neurons in comparison to the inhibitory presynaptic terminals in the dentate nucleus in human *post-mortem* cerebellar tissues derived from patients with mitochondrial disease[11].

To establish whether the mouse model recapitulates this neuropathological finding, we measured NDUFA13 expression normalised to SDHA in conjunction with the inhibitory cell marker GAD1-2 (Fig. 3a), which was employed to identify the inhibitory axons and presynaptic terminals of Purkinje neurons in the DCN.

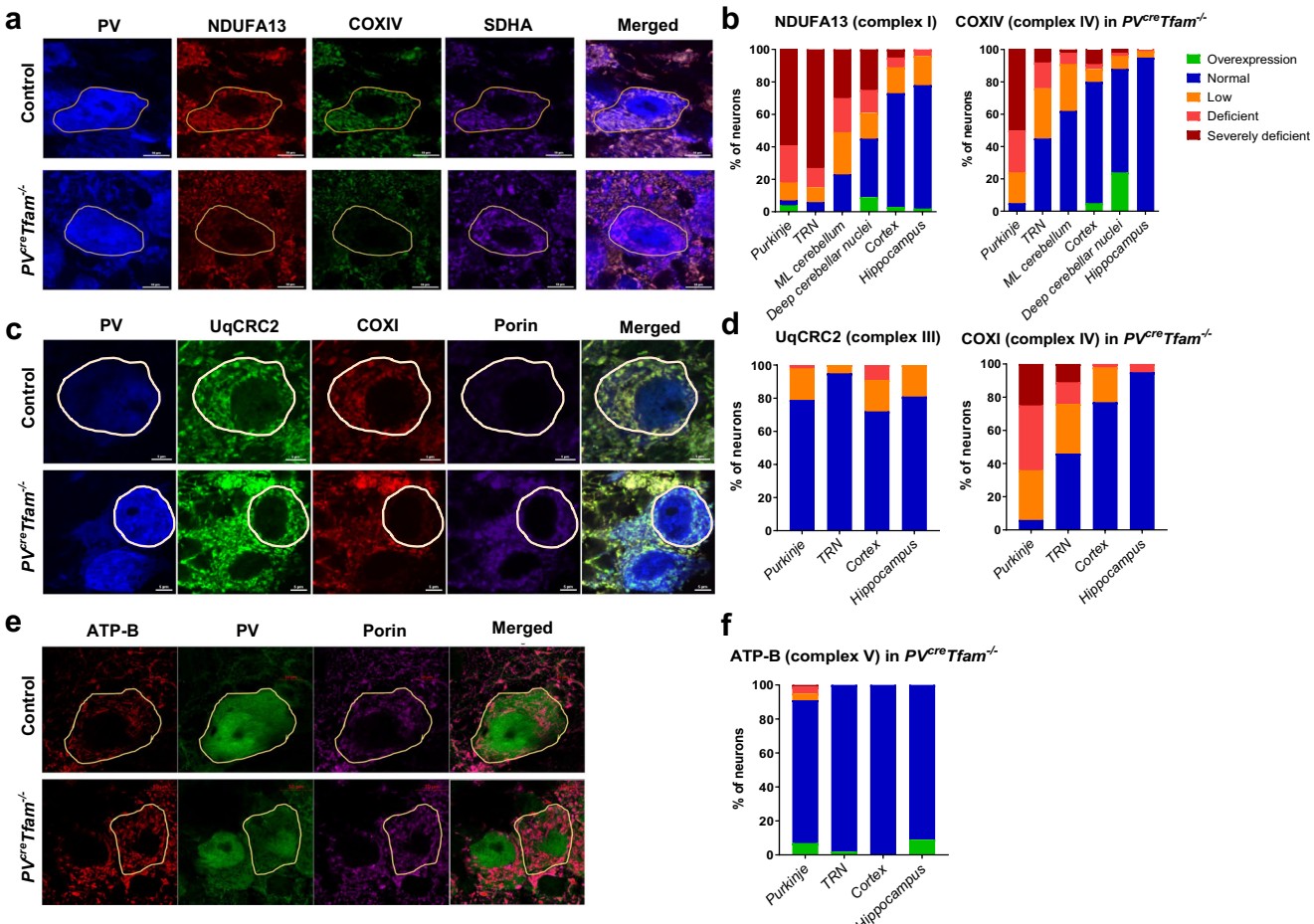

**Fig. 2 OXPHOS-functioning protein deficiencies in PV⁺ neurons in *PVᶜʳᵉTfam⁻/⁻* mice are brain region-dependent. a** Example quadruple confocal micrographs showing Purkinje cells with severe loss of NDUFA13 (complex I) and COXIV (complex IV) with intact SDHA (complex II, used as a mitochondrial mass marker) in the knockout mice. Scale bars – 10 μm. **b** Percentage of neurons in regions of interest with overexpressed (green), normal (blue), low (yellow), deficient (orange), and severely deficient (red) complexes I and IV subunit expression in the knockout mouse sections. Purkinje neurons were analysed in knockout animals ($n = 276$ neurons from 6 mice) and controls ($n = 130$ neurons from 3 mice); PV⁺ neurons of the TRN in knockout animals ($n = 282$ neurons from 6 mice) and controls ($n = 119$ neurons from 3 mice); PV⁺ interneurons of the molecular layer of the cerebellum in knockout animals ($n = 224$ neurons from 6 mice) and controls ($n = 129$ neurons from 4 mice); PV⁺ interneurons in somatosensory and visual cortical areas in knockout animals ($n = 110$ neurons from 6 mice) and controls ($n = 47$ neurons from 2 mice); PV⁺ interneurons across the hippocampal formation in the knockout mice ($n = 55$ neurons; 6 mice) and controls ($n = 20$ neurons; 3 mice). **c** Quadruple confocal images depicting PV, UqCRC2 (complex III), COXI (complex IV) and porin (mitochondrial mass). Scale bars – 5 μm. **d** Purkinje neurons were analysed in the knockout animals ($n = 212$ neurons from 6 mice) and controls ($n = 122$ neurons from 4 mice); PV⁺ neurons of the TRN in the knockout animals ($n = 171$ neurons from 5 mice) and controls ($n = 111$ neurons from 3 mice); PV⁺ interneurons in the somatosensory and visual cortical areas in the knockout animals ($n = 57$ neurons from 5 mice) and controls ($n = 43$ neurons from 3 mice); PV⁺ interneurons in the hippocampus in the knockout animals ($n = 21$ neurons from 5 mice) and controls ($n = 16$ neurons from 3 mice). **e** Triplex confocal images depicting ATP-B (complex V), PV and porin. Scale bars – 10 μm. **f** Purkinje neurons were analysed in the knockout animals ($n = 205$ neurons from 6 mice) and controls ($n = 140$ neurons from 5 mice); PV⁺ neurons of the TRN in the knockout animals ($n = 167$ neurons from 4 mice) and controls ($n = 176$ neurons from 4 mice); PV⁺ interneurons in the somatosensory and visual cortical areas in the knockout animals ($n = 57$ neurons from 3 mice) and controls ($n = 69$ neurons from 3 mice); PV⁺ interneurons in the hippocampus in the knockout animals ($n = 23$ neurons from 4 mice) and controls ($n = 25$ neurons from 3 mice).

Strikingly, despite almost all Purkinje cell bodies in the knockout mice displaying complex I decreased expression (Fig. 2b), Purkinje neuron presynaptic terminals in the DCN had a much milder decrease in NDUFA13 expression, with most inhibitory terminals exhibiting a detectable NDUFA13 signal (Fig. 3a). A more detailed interrogation of derived *z*-scores revealed that in GAD-immunoreactive terminals, median NDUFA13 expression was −1.753 in the knockout mice (Fig. 3b), which lies within the normal expression range, and is 2.5-fold higher in comparison to the Purkinje neuronal cell bodies, which was −4.425, and classed as severely deficient (Supplementary Fig. 3a). Moreover, it was established that GABA-synthesising

enzyme expression was diminished within the inhibitory axonal terminals surrounding the DCN neurons in the knockout group, with 24.8% of terminals exhibiting *z*-score of < −2, although it did not reach significance (Fig. 3c).

**PGC1 expression is unaltered in Purkinje and TRN neurons in the knockout mice.** To delineate whether mitochondrial biogenesis could be induced due to *Tfam* knockout in PV⁺ neurons, triplex immunofluorescence was utilised in which peroxisome proliferator-activated receptor gamma coactivator 1 (PGC1) expression was measured (Supplementary Fig. 7a). PGC1 acts

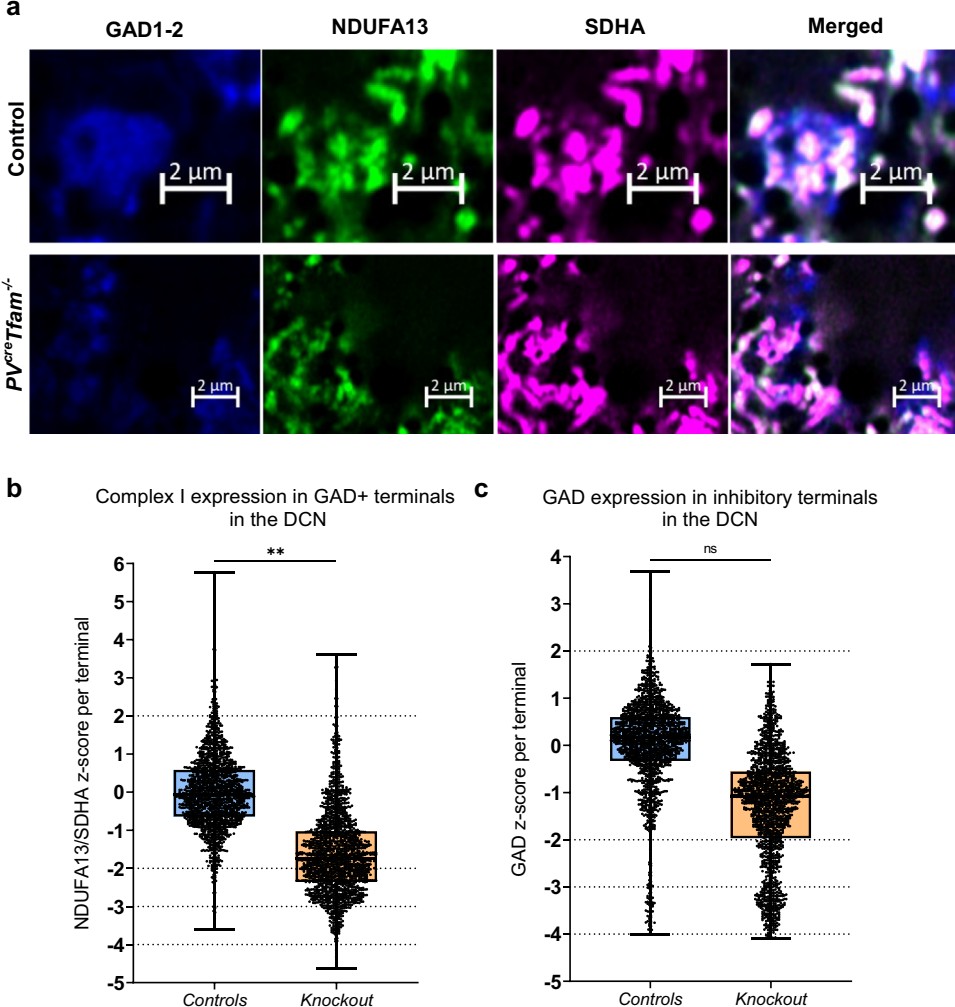

**Fig. 3 Milder complex I deficiencies in GABAergic inhibitory axonal terminals surrounding deep cerebellar nuclei neurons in comparison to Purkinje neuron cell bodies. a** Example confocal micrographs demonstrating GAD1-2 (blue), NDUFA13 (green) and SDHA (purple) immunofluorescence of inhibitory synapses surrounding DCN neurons. Knockout section demonstrates a mild reduction in NDUFA13 signal in comparison to the control, whereas SDHA expression is intact. Scale bars −2 μm. **b** Boxplot demonstrating z-score data NDUFA13/SDHA distribution in GAD-positive inhibitory terminals within DCN ($P = 0.0045$, linear mixed-effects model analysis; $n = 1594$ and 1676 terminals analysed from 4 mice per group). Dashed lines represent z-scores at normal ($2 < z < -2$), low ($< -2$), deficient ($< -3$), and severely deficient ($< -4$). **c** Boxplot demonstrating z-score GAD1-2 distribution in the inhibitory terminals within DCN ($P = 0.0739$, linear mixed-effects model analysis; $n = 4$ mice per group).

upstream of TFAM and is involved in mitochondrial biogenesis and inhibition of mitochondrial fission[31]. Neither Purkinje nor TRN neurons exhibited a statistically significant change in PGC1 or porin levels (Supplementary Fig. 7b–e).

**Anaplerosis enzyme pyruvate carboxylase is overexpressed in a proportion of Purkinje and TRN neurons in the knockout mice.** Purkinje neurons demonstrated the greatest level of severe combined respiratory chain deficiencies in complexes I and IV. A recent study by Motori and colleagues[32] demonstrated that Purkinje neurons can be protected from neurodegeneration due to severe OXPHOS defects by increasing the expression of anaplerosis enzymes that replenish substrates for tricarboxylic acid (TCA) cycle, such as malic enzyme and pyruvate carboxylase (PC), which converts pyruvate to oxaloacetate.

We assessed PC levels in individual Purkinje neurons (Fig. 4a) and found an increase in the median optical density (Fig. 4b), despite not reaching a statistical significance when analysed per mouse ($P = 0.0681$). Over 55% of Purkinje neurons from knockout mice exhibited a z-score of > 2 for PC (Fig. 4b),

suggesting that metabolic remodelling is taking place in OXPHOS-deficient Purkinje neurons. Similarly, 42% of all TRN neurons from knockout mice demonstrated overexpression of PC (Fig. 4c), despite not reaching statistical significance when analysed on a group level per mouse. No significant differences were detected in pyruvate carboxylase expression in the cortex or hippocampus of the knockout mice (Supplementary Fig. 8).

**Ectopic expression of tyrosine hydroxylase and c-Fos immunoreactivity in Purkinje neurons in the knockout mice indicate abnormal Ca²⁺ handling and increased neuronal activity.** We hypothesised that OXPHOS deficiency detected in Purkinje neurons may lead to intracellular $Ca^{2+}$ buffering dysregulation due to mitochondrial dysfunction. To establish whether $Ca^{2+}$ overloading occurs within Purkinje neurons as a result, immunofluorescence was used to measure the optical density of ectopic tyrosine hydroxylase expression (Fig. 4d), which typically arises due to excessive intracellular $Ca^{2+}$ levels and when excitation is altered[33,34]. The z-scores indicated that 15% of Purkinje neurons demonstrated overexpression in tyrosine hydroxylase in the

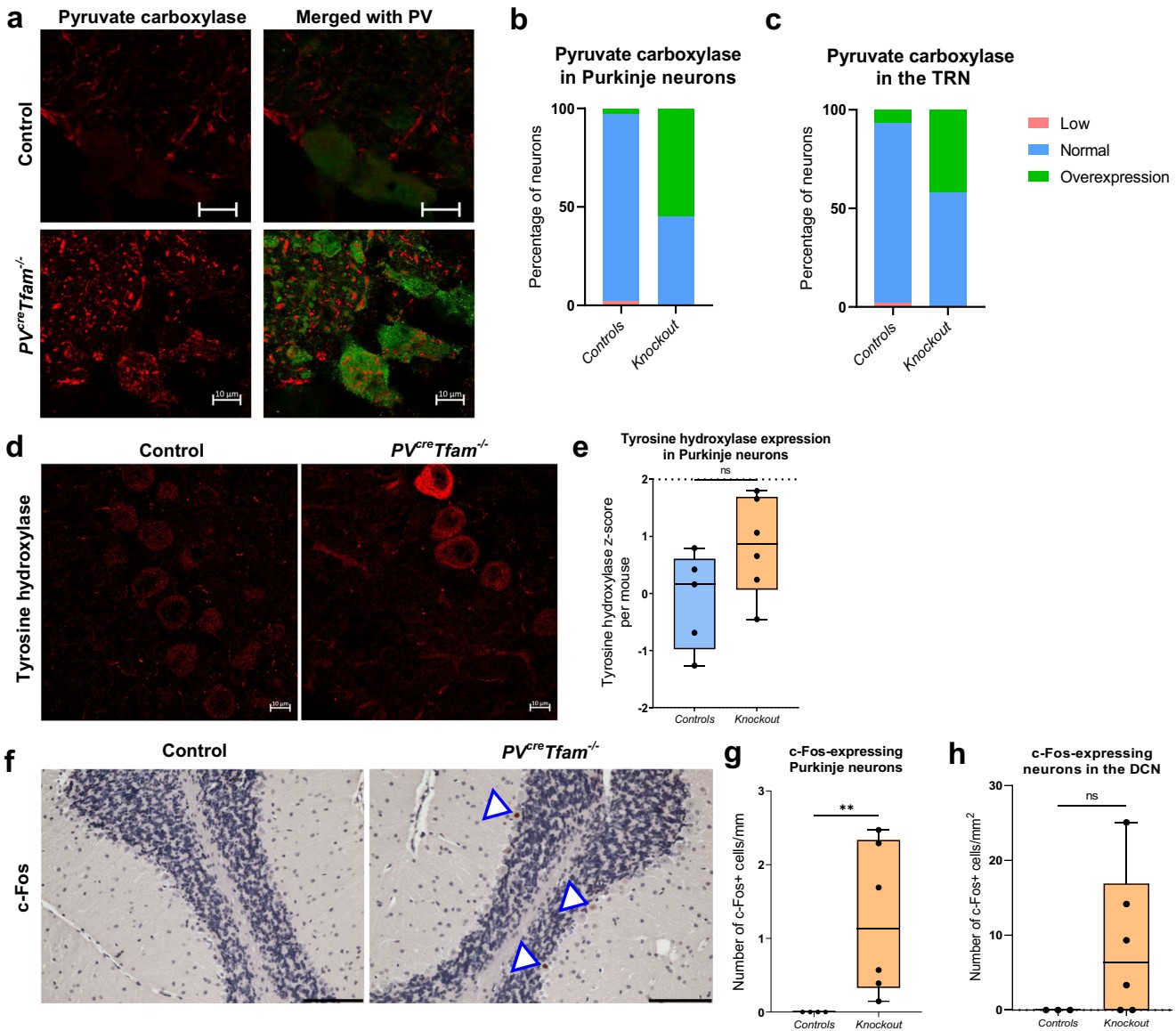

**Fig. 4 Metabolic remodelling via pyruvate carboxylase expression, accompanied by signs of hyperexcitability such as ectopic expression of tyrosine hydroxylase and c-Fos expression in Purkinje neurons of the knockout mice. a** Example confocal micrographs of Purkinje neurons (PV, green) and pyruvate carboxylase (PC) co-staining (red) in littermate control and knockout mouse cerebellum. Scale bars – 10 μm. **b** Increase in the percentage of Purkinje neurons displaying overexpression of PC. $P = 0.0681$, linear mixed-effects model; $n = 107$ neurons from 3 control and $n = 230$ from 6 knockout mice. **c** Increase in the percentage of PV+ neurons of the TRN with PC overexpression in the knockout mice. $P = 0.5743$, linear mixed-effects model; $n = 54$ neurons from 2 control and $n = 108$ neurons from 3 knockout mice. **d** Example confocal micrographs of Purkinje neurons expressing tyrosine hydroxylase. Scale bars – 10 μm. **e** Quantification of optical density $z$-scores of tyrosine hydroxylase in individual Purkinje neurons revealed a non-significant increase in the knockout mouse group ($P = 0.1022$, linear mixed-effects model; $n = 166$ neurons from 5 control and $n = 228$ from 6 knockout mice). **f** Example light micrograph of c-Fos immunohistochemical staining in Purkinje neurons in the knockout cerebellum. Arrowheads show neurons with positive c-Fos signal detected within nuclei. Scale bar – 100 μm. **g** Graph demonstrates the number of c-Fos-immunoreactive Purkinje cells per unit length of the Purkinje cell layer ($n = 4$ control and $n = 6$ knockout mice). **h** c-Fos-expressing neuronal density in the DCN was non-significantly increased in the knockout group ($P = 0.1667$, Mann–Whitney test; $n = 3$ control and $n = 6$ knockout mice).

knockout mice, although the difference between the groups did not reach significance when data were analysed per mouse (Fig. 4e).

To gain insights into the effects on neuronal activity levels and potential excitation-inhibition disturbances, c-Fos was employed as a surrogate neuronal activity marker using immunohistochemistry[35] (Fig. 4f). The number of c-Fos expressing Purkinje neurons was significantly greater in the knockout mice (Fig. 4g), indicating potential hyper-excitation of Purkinje neurons, while no c-Fos-immunoreactive Purkinje neurons were detected in control mice.

Similarly, this change was mirrored by a comparable degree of increase in the density of c-Fos-immunoreactive DCN neurons, although it did not reach significance (Fig. 4h), indicating that both Purkinje and DCN neurons might be more active in the knockout mice, signifying a possibility of reduced inhibition within the cerebellar loop.

**No prominent neurodegeneration of PV+ neurons despite the variability in the OXPHOS deficiencies.** PV+ Purkinje neurons were found to be most affected in the knockout mice by

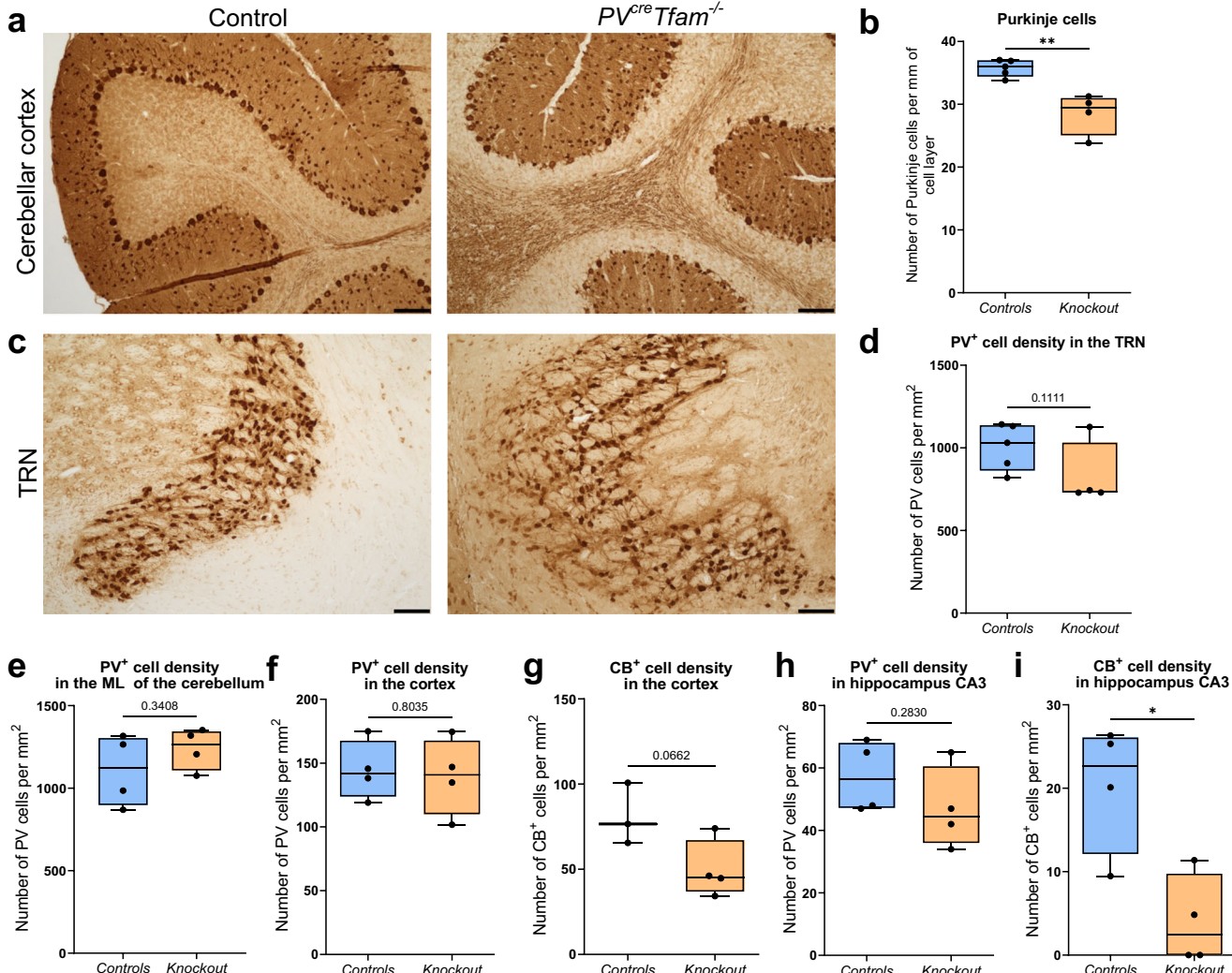

**Fig. 5 Changes in PV$^+$ neuronal density according to the brain region. a** Immunohistochemistry of PV in the cerebellum demonstrating a loss of PV$^+$ Purkinje cells in the knockout mice. **b** Number of PV-immunoreactive Purkinje neurons per length of the Purkinje cell layer (cells were only analysed if arranged in a single lamina) was significantly lower in the knockout mice ($n = 4$) in comparison to control ($n = 5$) mice ($P = 0.0026$, $t$-test). **c** Immunohistochemistry of PV neurons n the thalamic reticular nucleus (TRN) and **d** quantified PV$^+$ cell density in the TRN ($n = 5$ control and $n = 4$ knockout mice). **e** Density of PV$^+$ interneurons within the molecular layer of the cerebellum ($n = 5$ control and $n = 4$ knockout mice). **f** PV$^+$ ($n = 4$ control and $n = 4$ knockout mice) and **g** CB$^+$ interneuron density estimation in the somatosensory and visual cortical areas ($n = 3$ control and $n = 4$ knockout mice). **h** PV$^+$ interneuron densities and **i** CB$^+$ interneuron loss in the CA3 area in the hippocampus in the knockout mice ($P = 0.283$ and $0.0136$, respectively, $t$-test; $n = 4$ mice per group). All scale bars — 100 μm.

neurodegeneration (Fig. 5a). A mild 20% reduction in the mean number of PV$^+$ Purkinje neurons from 35.8 neurons per mm of Purkinje cell layer length in control mice to 28.5 in the knockout mice was detected (Fig. 5b). The mean density of PV$^+$ neurons in the TRN (Fig. 5c) was decreased by approximately 20% as well, however without reaching the statistical significance level (Fig. 5d, $P = 0.1111$).

There were no differences in the PV$^+$ interneuron density within the molecular layer of the cerebellum (Fig. 5e, $P = 0.3408$) or cortical PV$^+$ interneurons (Fig. 5f, $P = 0.8035$). Surprisingly, a trend suggesting a decrease in the density of calbindin-expressing (CB$^+$) interneurons was detected in the somatosensory and visual cortical regions (Fig. 5g, $P = 0.0662$).

The mean density of PV$^+$ interneurons within the CA3 subregion of the hippocampus was 47 cells per mm$^2$, which was lower in comparison to the littermate control mean density of 57 cells per mm$^2$, albeit not reaching statistical significance (Fig. 5h, $P = 0.2830$). Similar to the posterior cortical regions, the

hippocampal CA3 region also displayed a diminished density of CB$^+$ interneurons in the knockout animals, which was statistically significant (Fig. 5i).

**Microglial and astrocytic involvement in the cerebellum.** Since the cerebellum demonstrated a reduction in PV$^+$ Purkinje neuron density and extensive OXPHOS deficiencies, we hypothesised that it may lead to secondary changes in glial cells. Therefore, we investigated glial cell changes in the cerebellar cortex and DCN.

In control mice, microglia appeared to have small somata and ramified long processes suggestive of the quiescent state (Fig. 6a, c). In contrast, microglia in the knockout mice displayed reactive morphology, characterised by the shortening and thickening of the processes as well as the enlargement of the cell bodies (Fig. 6a, c). The density of Iba-1-expressing microglia and/or macrophages was significantly elevated in the cerebellar cortex and DCN of knockout mice, with DCN demonstrating a slightly greater overall increase in the density (Fig. 6b, d).

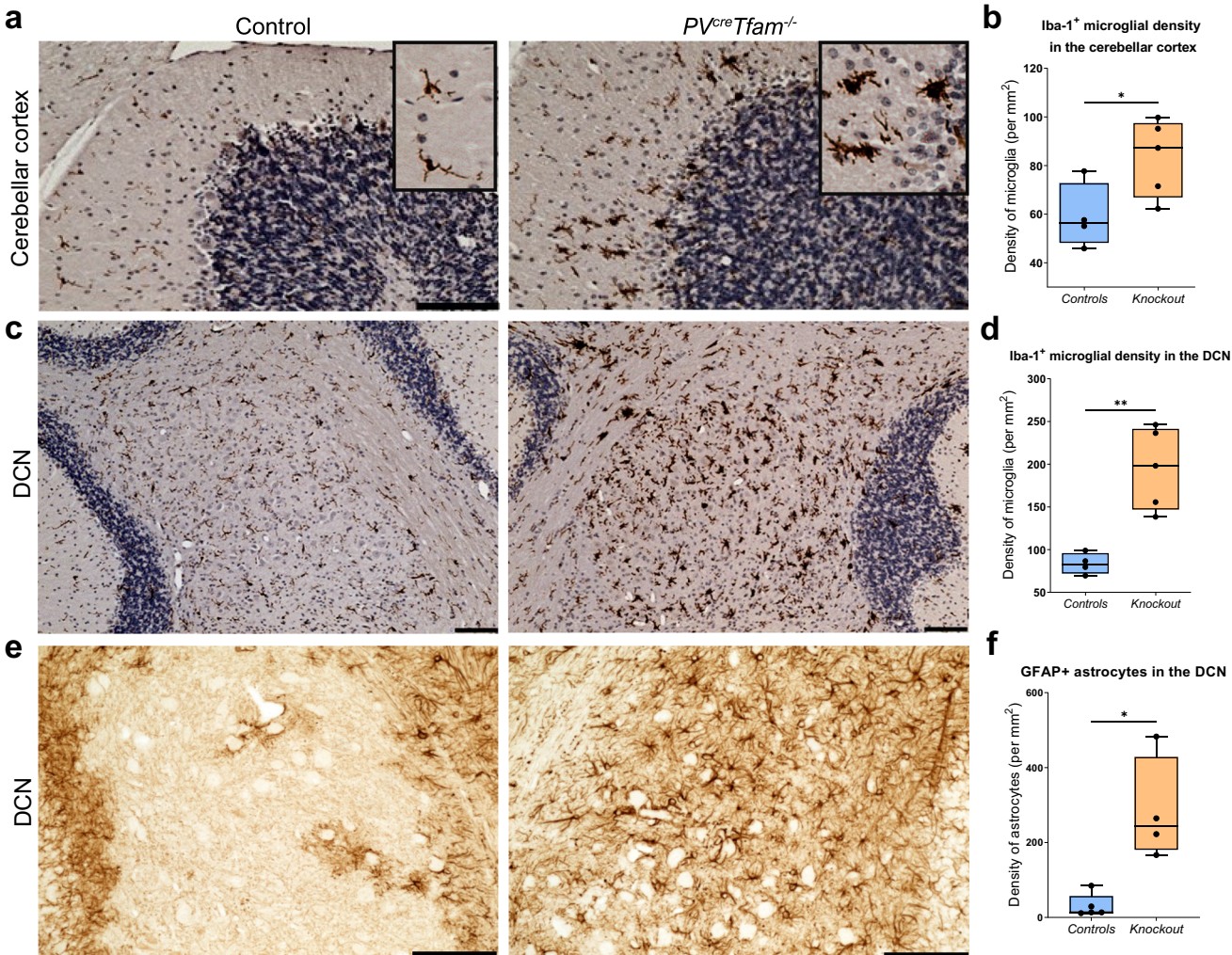

**Fig. 6 Severe OXPHOS deficiencies in PV⁺ neurons of the cerebellum are sufficient to trigger secondary microglial and astrocytic changes. a** Light micrographs depict microglial activation in the cerebellar cortex of knockout animals. Inserts show microglia/macrophages in the molecular layer of the cerebellar cortex with distinct morphology between the groups: ramified and elongated processes in the littermate control group suggestive of quiescent microglia and amoeboid, enlarged cell bodies with retracted processes in the knockout group suggestive of microglial reactivity. Scale bars – 100 µm. **b** Mean density of Iba-1⁺ microglia/macrophages in the cerebellar cortex of the knockout animals was significantly greater than in controls ($P = 0.0463$, $t$-test; $n = 4$ control and $n = 5$ knockout mice). **c** Microglial activation in the DCN of knockout animals. Scale bars –100 µm. **d** Mean density of Iba-1⁺ microglia/macrophages in the DCN of the knockout animals was greater than in controls ($P = 0.0028$, $t$-test; $n = 4$ control and $n = 5$ knockout mice). **e** GFAP immunohistochemistry depicting reactive astrocytes in the DCN in control and knockout mice. Scale bars –100 µm. **f** Median density of reactive astrocytes was significantly increased in the DCN region ($P = 0.0159$, Mann–Whitney test; $n = 5$ control and $n = 4$ knockout mice).

Visual assessment of astrocytic reactivity confirmed the presence of Bergmann gliosis around Purkinje cells and within the molecular cell layer in the knockout mice. A marked increase in the median number of GFAP⁺ astrocytes, a marker of reactive astrocytes, was detected in knockout mice DCN compared to control DCN (Fig. 6e, f).

Microglia/macrophages in the cerebellum in the knockout group were found to have an increased soma area size. The mean soma size in the knockout group was 51 µm², which was significantly greater than in the littermate control group which was 34 µm² (Fig. 7a). Moreover, microglial processes were significantly shorter in the knockout group. The mean microglial process length in the knockout group was 20.81 µm, which was 20% shorter than the littermate controls median process length of 26.14 µm (Fig. 7b). The percentage of microglia contacting Purkinje neurons was doubled in the knockout group in comparison to the littermate controls, with almost 12% in the knockout group, in contrast to only 5.6% in the control

cerebellum and DCN, however, the data per mouse were not significant on the group level (Fig. 7c). Three-dimensional analysis of the z-stack images of microglia in the cerebellar cortex and DCN confirmed the two-dimensional data described above, with microglia in the knockout group displaying a statistically significant increase in their mean volume in both regions (Fig. 7d, e).

**OXPHOS deficiencies in PV⁺ interneurons in the human primary visual cortex.** To validate the findings of the mouse model, we measured OXPHOS deficiency levels specifically in PV⁺ interneurons in *post-mortem* tissues from patients with mitochondrial disease. The cohort of patients included in this study (Table 1) presented with neurological manifestations, including ataxia ($n = 9$), epilepsy ($n = 7$), and stroke-like episodes ($n = 4$).

Human *post-mortem* tissues from the primary visual cortex located in the occipital lobe (Brodmann area 17, or BA17) were

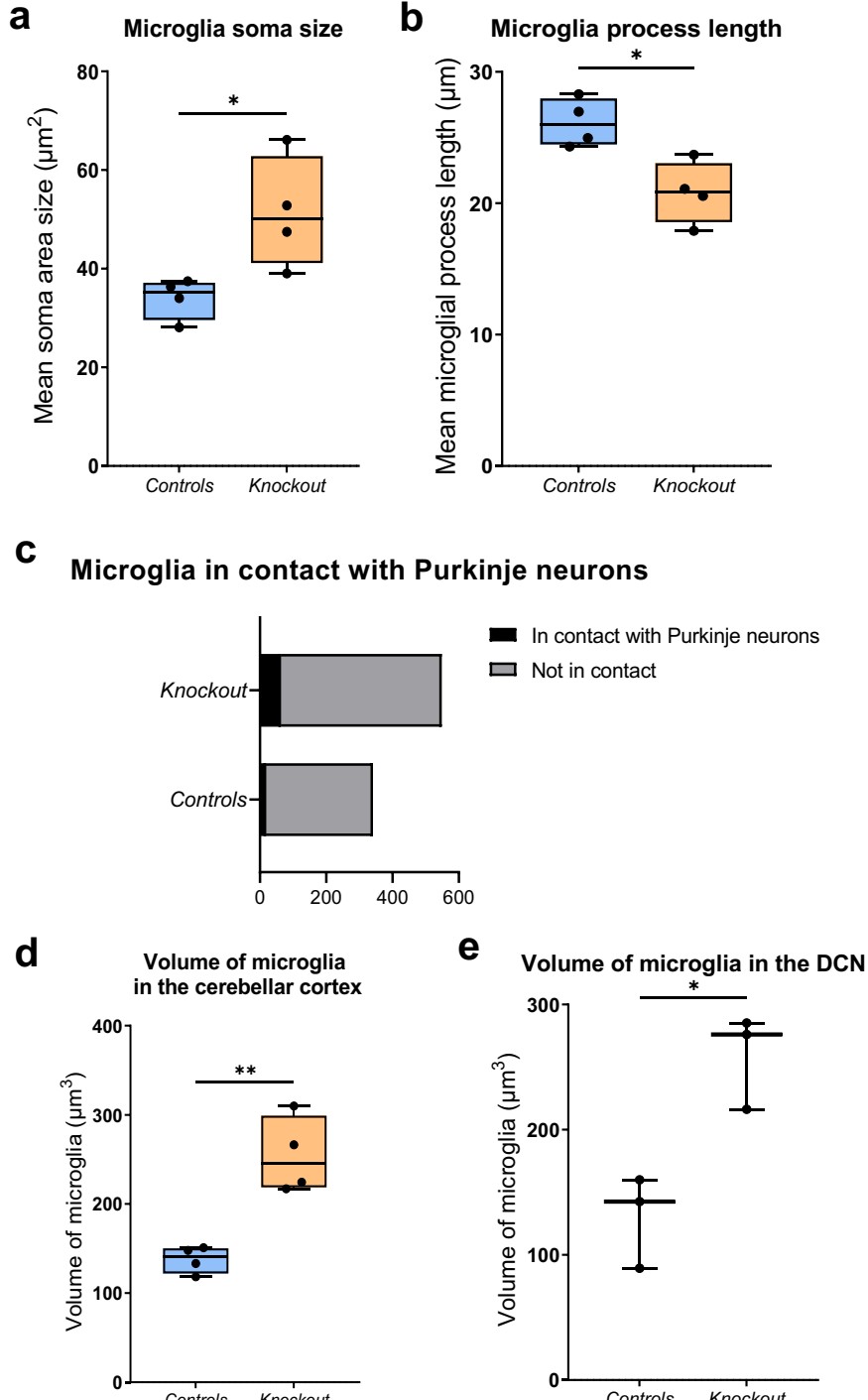

**Fig. 7 Microglia display a reactive phenotype in the cerebellum and a greater proportion of Purkinje neuron contacts in the knockout mice. a** Microglial soma size is significantly increased in the knockout group ($P = 0.0281$, $t$-test; $n = 341$ microglia from 4 control mice and $n = 549$ from 5 knockout mice). **b** Microglial process length is shorter in the knockout group ($P = 0.0121$, $t$-test). **c** Absolute number of microglia in contact with Purkinje neurons (black) and not in contact (grey) within the cerebellum. The proportion of microglia displaying contact with Purkinje cells is increased in the knockout mice, although it did not reach significance ($P = 0.1111$, linear mixed-effects model analysis; $n = 4$ control and $n = 5$ knockout mice). **d** Mean microglial volume is significantly increased in the knockout group in the cerebellar cortex ($P = 0.0021$, $t$-test; $n = 61$ microglia from 4 control mice and $n = 86$ from 4 knockout mice) and **e** DCN ($P = 0.0132$, $t$-test; $n = 15$ microglia from 3 control mice and $n = 34$ from 3 knockout mice).

selected due to the highest density of PV$^+$ neurons in this region in controls in comparison to frontal and temporal lobes[15]. Furthermore, stroke-like episodes and epileptic activity have been documented in the occipital lobe in mitochondrial disease patients[6]. Previously, profound OXPHOS deficiencies were reported in GABAergic interneurons in the primary visual cortex

relative to frontal and temporal cortices in patients with mitochondrial disease[12].

We identified that in control tissues, the total expression of porin, complex I subunit NADH: ubiquinone oxidoreductase B8 (NDUFB8) and complex IV subunit COXI were significantly greater in PV$^+$ interneurons compared to non-PV-expressing cells (Fig. 8).

**Table 1 Summary of clinical symptoms, genetic diagnoses, and demographic information of patients with mitochondrial disease.**

| Patient | Genetic diagnosis | Heteroplasmy | Sex | Age at death | Clinical features | Cause of death |
|---|---|---|---|---|---|---|
| Patient 1 | m.3243 A > G | Muscle: 85% | M | 45 | MELAS, epilepsy, cognitive impairment, ataxia, deafness, HCM | N/A |
| Patient 2 | m.3243 A > G | Muscle: 77% | F | 64 | MELAS, epilepsy, cognitive impairment, ataxia, deafness, diabetes | Terminal stage of MELAS syndrome |
| Patient 3 | m.3243 A > G | Muscle: 83% | M | 54 | MELAS, epilepsy, cognitive impairment, ataxia, deafness, diabetes, intestinal pseudo-obstruction | Sepsis |
| Patient 4 | m.3243 A > G | Muscle: 53% | M | 61 | Recurrent encephalopathy, ataxia, brainstem ischaemic stroke | Aspiration pneumonia |
| Patient 5 | m.3243 A > G | Muscle: 71% | M | 30 | HCM, deafness | Heart failure |
| Patient 6 | m.8344 A > G | Blood: 92% | F | 42 | MERRF, myoclonic epilepsy, cognitive impairment, ataxia, peripheral neuropathy | Respiratory failure |
| Patient 7 | m.8344 A > G | Urine: 70% | M | 58 | MERRF/MELAS overlap syndrome, myoclonic epilepsy, cognitive impairment, ataxia, peripheral neuropathy | Stroke-like episodes |
| Patient 8 | m.8344 A > G | Muscle: 98% | M | 31 | MERRF, myoclonic epilepsy, ataxia, dysphagia, mild LVH | Respiratory failure |
| Patient 9 | POLG (p.Gly848Ser and p.Ser1104Cys) | - | M | 59 | Ataxia, parkinsonism, CPEO, peripheral neuropathy | Suppurative tracheobronchitis |
| Patient 10 | POLG (p.Thr251Ile/p.Pro587Leu and p.Ala467Thr) | - | M | 79 | Ataxia, cognitive impairment, CPEO | Pneumonia |
| Patient 11 | POLG (p.Trp748Ser, p.Arg1096Cys and polymorphic variant p.Glu1143Gly) | - | M | 55 | Epilepsy, myoclonus, cognitive impairment, ataxia, tremor, CPEO, peripheral neuropathy | Lower respiratory tract infection |

*MELAS* mitochondrial encephalopathy with lactic-acidosis and stroke-like episodes, *MERRF* myoclonic epilepsy with ragged-red fibres, *CPEO* chronic progressive external ophthalmoplegia, *HCM* hypertrophic cardiac myopathy, *LVH* left ventricular hypertrophy.

In total, 775 PV$^+$ neurons from 16 age-matched controls and 469 neurons PV$^+$ neurons from 11 mitochondrial disease patients were analysed by quantifying OXPHOS subunits normalised for mitochondrial content as assessed by porin. Patients demonstrated a greater percentage of PV$^+$ neurons classed as low, deficient, or severely deficient in terms of OXPHOS expression using derived $z$-scores, with a high degree of variability between patients (Fig. 9a, b). NDUFB8 expression normalised to porin (Fig. 9a), was more severely affected than that of COXI normalised to porin (Fig. 9b). Moreover, an increase in the expression of porin was observed in PV$^+$ interneurons in patients with mitochondrial disease, albeit without reaching a statistical significance (Fig. 9c).

Neurodegeneration of PV$^+$ interneurons in the mitochondrial disease patient cohort was also evident (Fig. 9d). The mean density of PV$^+$ interneurons in the BA17 region in the mitochondrial disease group was significantly decreased by approximately 30% in comparison to age-matched controls (Fig. 9e).

Interestingly, corroborating the data reported in a rare paediatric mitochondrial disease Alpers' syndrome[15], we also observed a non-significant decrease in PV expression in surviving PV$^+$ interneurons of the primary visual cortex in adult patients with mitochondrial disease (Supplementary Fig. 9).

## Discussion

In this study, a novel murine model was devised, which harbours mitochondrial dysfunction in parvalbumin-expressing cells, achieved via mtDNA depletion in vivo, recapitulating major neurological impairments observed in mitochondrial disease. We demonstrated that $PV^{cre}Tfam^{-/-}$ mice developed a juvenile-onset progressive phenotype characterised by reduced weight, anxiety, cognitive impairment, seizure-like phenotype, and severe ataxia, eventually resulting in a shortened lifespan. The juvenile-onset, and the rapid and progressive nature of the neurological symptoms in this mouse model, render it suitable for future preclinical studies in mitochondrial disease as well as other types of epilepsy or neuropsychiatric disorders.

At 10 weeks of age, mice developed spontaneous hyperlocomotion assessed by the open-field test; head-nodding phenotype, which could be a manifestation of either epilepsy or tremor; stargazing episodes, characteristic of absence-like seizures in rodents[26], as documented in the *stargazer* mouse model, which displays stargazing seizures and ataxia[26]. The stargazing phenotype is thought to arise due to thalamic reticular nucleus (TRN) impairment[26]. Genetic models of TRN disinhibition via GABAergic impairment demonstrated that normal activity of TRN is crucial for preventing abnormal thalamocortical hypersynchrony, which may result in generalised absence epilepsy if TRN is overactivated[36,37]. In a transgenic *Gria4* knockout mouse model, where there was insufficient excitation of TRN by cortico-TRN projections, thalamocortical neurons were overexcited via corticothalamic inputs, and absence-like seizures were noted[38]. Additionally, a computational modelling study proposed that a loss of thalamic feed-forward inhibition may precipitate absence seizures[39]. Our study shows a decrease in the overall TRN PV$^+$ neuron density in the knockout mice, accompanied by profound complex I deficiency in almost all surviving neurons and 60% of those also exhibited decreased or deficient expression of complex IV, which may cause the absence-like seizures in this mouse model.

Severe motor deterioration led to ataxia, which was characterised by reduced maximal speed of rotation on the rod, at which mice lost coordination and their reduced latency to fall in the rotarod test, splayed hindlimbs, and abnormal gait at

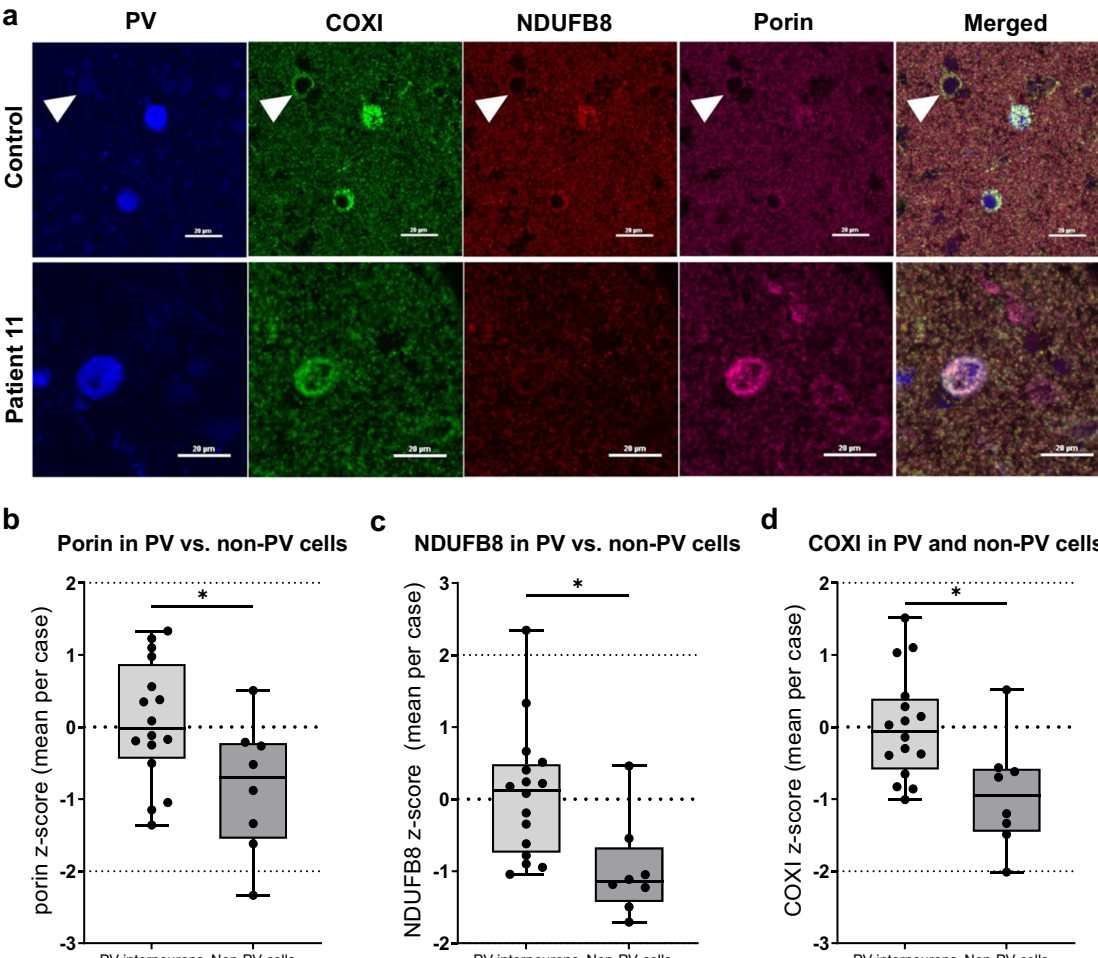

**Fig. 8 PV⁺ interneurons of the primary visual cortex display a greater overall expression of mitochondrial mass, complex I and complex IV subunits in comparison to non-PV-expressing cells in control tissue and patients with mitochondrial disease frequently demonstrate a decrease in complex I subunit expression. a** Example confocal micrographs demonstrating PV (blue), COXI (green), NDUFB8 (red) and porin (purple) immunofluorescence in primary visual cortex in control and patient 11 (*POLG*). Arrowhead points to a non-PV-immunoreactive cell that appears to have reduced mitochondrial mass, NDUFB8 and COXI expression in relation to PV⁺ interneurons in control tissues. Patient 11 PV⁺ interneuron shows almost complete loss of NDUFB8, decreased COXI, and increased porin signal. Scale bars – 20 μm. **b–d** Boxplots demonstrating significantly increased porin, NDUFB8 and COXI (not normalised to porin) expression in PV⁺ interneurons vs. non-PV-immunoreactive cells in the occipital lobe of neurologically normal controls ($P = 0.0235$, 0.0131, 0.0131, respectively; *t*-test with Benjamini–Hochberg adjustment; $n = 775$ PV⁺ interneurons from 16 controls and $n = 453$ non-PV-immunoreactive cells analysed from 8 controls).

12–13 weeks of age. Ataxia and motor deterioration constituted the most prominent phenotypic features of the knockout mice, and neuropathological examination corroborated the phenotypic features by highlighting the cerebellum as the most affected region investigated. Specifically, we report PV⁺ Purkinje neuron loss, with surviving Purkinje neurons exhibiting combined complex I and IV severe deficiencies, resembling neuropathological findings reported in human *post-mortem* tissues from adult and paediatric patients with these mitochondrial diseases[9,11,13]. We speculate that the disparity in OXPHOS protein defects among PV⁺ neurons is in part due to Purkinje neurons inherent high firing rates and tonic mode of spiking[40] and large cell size[41], which may render this energy-demanding neuronal subtype particularly vulnerable to mitochondrial impairment. In response to OXPHOS deficiency, a proportion of PV⁺ neurons showed overexpression of pyruvate carboxylase in the cerebellum and TRN, which suggests metabolic remodelling to support anaplerosis.

Moreover, the surviving Purkinje neurons demonstrated signs of hyperexcitability with enhanced expression of c-Fos which was not seen in control littermates' cerebellum, and was accompanied by a similar increase in the number of DCN c-Fos-immunoreactive neurons. We also report ectopic overexpression of tyrosine hydroxylase in a proportion of the knockout mice Purkinje neurons. Tyrosine hydroxylase expression is induced in Purkinje cells upon a sustained rise in the level of intracellular $[Ca^{2+}]$[33,34]. The accumulation of intracellular $Ca^{2+}$ ions was hypothesised in Purkinje neurons in this mouse model due to established mitochondrial roles in buffering excess $Ca^{2+}$ ions during neuronal activity[42]. We speculate that Purkinje neuron mitochondrial dysfunction and subsequent $Ca^{2+}$ overload, coupled with a diminished expression of GABA-synthesising GAD enzyme in their axonal terminals, would have led to a perturbation in GABAergic neurotransmission and reduced inhibition of the DCN, which would explain an increase in c-Fos-positive DCN neurons and ataxia phenotype. Furthermore, cerebellar OXPHOS and neurodegenerative changes were accompanied by astrogliosis and microgliosis in the cerebellar cortex and deep cerebellar nuclei. Recent literature suggests that in a mouse model of Leigh syndrome characterised by *Ndufs4* complex I subunit knockout,

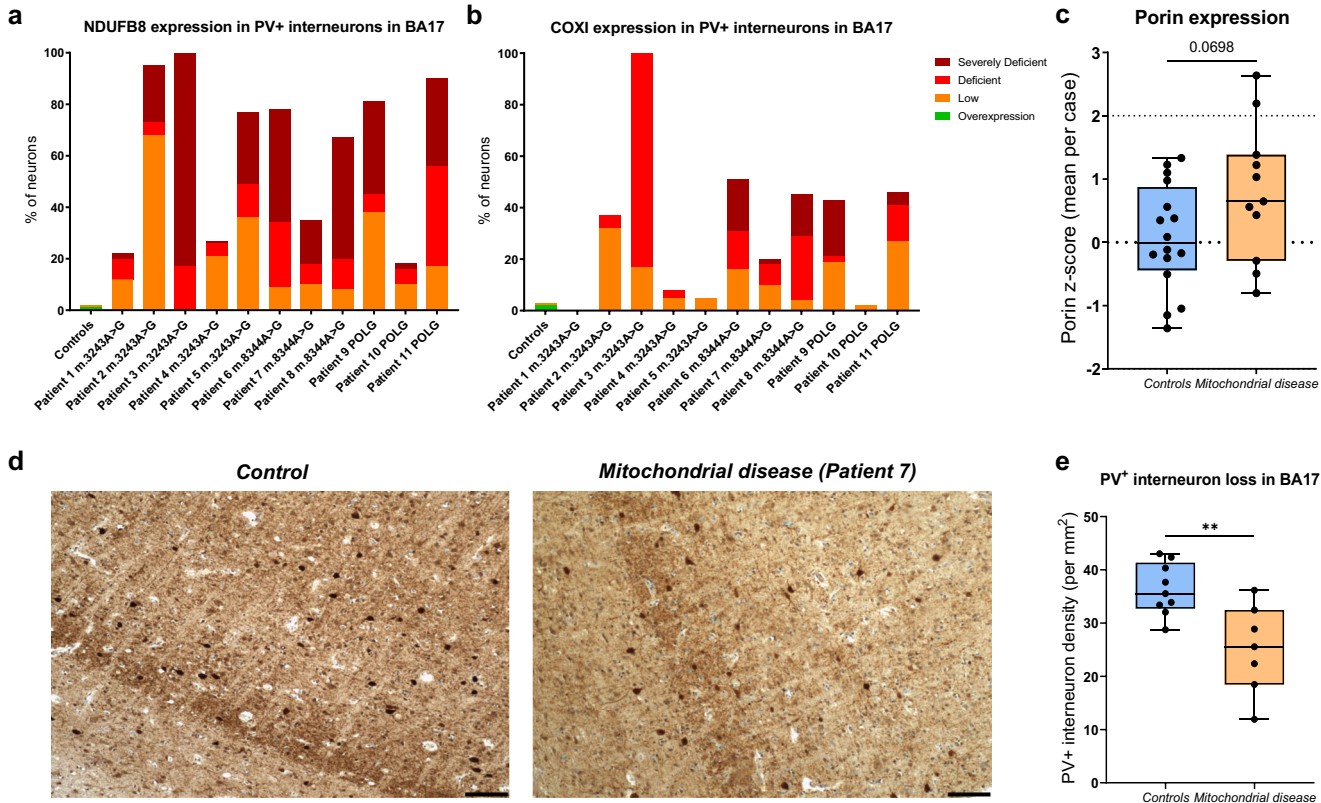

**Fig. 9 NDUFB8 (complex I) and COXI (complex IV) OXPHOS deficiencies detected in PV+ interneurons of the primary visual cortex in adult patients with mitochondrial disease. a** Bar chart demonstrates the percentage of neurons in each classification of normalised NDUFB8/porin expression in PV+ interneurons in the BA17 area. Coloured bars denote overexpression (green), low (orange), deficient (red) or severely deficient (dark red) expression. **b** Bar chart demonstrates the percentage of neurons in each patient of normalised COXI/porin expression in PV+ interneurons in the BA17 area in each group: low (orange), deficient (red) or severely deficient (dark red). In total, n = 775 neurons from 16 control subjects and n = 469 from 11 patients were analysed. **c** A statistical trend towards an increase in mean porin expression in PV+ interneurons in the mitochondrial disease group was observed (P = 0.0698, t-test). **d** Example images of BA17 area of the occipital cortex with PV+ interneurons in brown in control and mitochondrial disease patient 7. All scale bars ~100 μm. **e** PV+ interneuron cell density in controls (n = 9) and mitochondrial disease patients (n = 7, patients 1, 2, 3, 6, 7, 9, 11) with a statistically significant reduction in the density of PV+ interneurons in the mitochondrial disease group (P = 0.0044, t-test).

depletion of microglia resulted in amelioration of astrogliosis in the cerebellar cortex and improvement of motor coordination on the rotarod and prolonged lifespan[43], suggesting that chronic microglial pro-inflammatory changes could be detrimental.

In posterior cortical and hippocampal regions, we detected a decrease in the density of CB+ interneurons, despite the relative preservation of PV+ interneurons and mild OXPHOS deficiencies in these brain regions. The loss of CB+ interneurons from these regions validates the novel mouse model further as previous human *post-mortem* studies showed similar findings in patients with Alpers syndrome[15] and MELAS[44], and could suggest excitotoxic damage in these brain regions.

We performed a neuropathological study in the *post-mortem* human tissues derived from patients with primary mitochondrial disease, with ataxia and epilepsy being the most prevalent symptoms in this cohort. Previously published studies using human *post-mortem* tissues demonstrated severe extensive OXPHOS deficiencies and loss of GABAergic inhibitory interneurons in the occipital lobe in adult and paediatric patients with mitochondrial disease[12,13], with the profound vulnerability of the PV+ interneuron subclass to cell death and mitochondrial impairment in Alpers' syndrome[15]. This study focused on the PV+ interneurons in the occipital cortex of adult patients with genetically and clinically characterised mitochondrial disease. To assess mitochondrial respiratory chain complexes expression, antibodies against NDUFB8 subunit of complex I and the COXI

subunit of complex IV were used, which are known to be associated with complexes I and IV function, respectively[45,46]. Our study demonstrates a greater overall expression of mitochondrial mass as well as complex I and IV expression in PV+ interneurons in comparison to non-PV-expressing cells in control tissues, which provides further evidence for the PV+ energy hypothesis[14]. Patients with mitochondrial disease exhibited a more profound deficiency in complex I subunit expression, and a milder deficiency in complex IV subunit within the PV+ interneurons, irrespective of the pathogenic variant, in agreement with published data in cortical GABAergic interneurons[12] and Purkinje neurons of the cerebellum[9]. Literature suggests that complex I, but not complex IV, deficient neurons are more likely to withstand neurodegenerative processes and survive, and that complex IV deficiency levels may correlate with the degree of neuronal loss[9,47]. This study demonstrated that there is an overall reduction in the mean density of PV+ interneurons by approximately 30% in the mitochondrial disease adult patient cohort with primarily MELAS and MERRF syndromes, which is much milder than in Alpers' syndrome[15].

In conclusion, the novel mouse model of mitochondrial dysfunction in PV+ neurons reported here demonstrated epilepsy and ataxia and exhibited a severe phenotype without overt neuronal loss, suggesting that mitochondrial dysfunction in the absence of severe neurodegeneration is sufficient to induce detrimental neurological disorder in vivo, rendering it a valuable

model for preclinical testing of therapeutic agents, including pharmacological agents or neurostimulatory methods with great translational potential.

## Methods

**Generation of a transgenic mouse model.** $Tfam^{loxP}/Tfam^{loxP}$ mice of mixed genetic background that harbour two $loxP$ sites flanking exons 6 and 7 of the $Tfam$ gene[21,48] were mated to $+/PV^{cre}$ mice (Jackson Laboratory 008069). Compound heterozygotes ($+/Tfam^{loxP}$, $+/PV^{cre}$) were identified and crossed to generate mice with the genotype $+/PV^{cre} Tfam^{loxP}/Tfam^{loxP}$.

The resulting offspring demonstrated normal litter sizes and conformed to Mendelian distribution for $Tfam$ genotype (28.4% wildtype; 49% heterozygous; 22.5% homozygous; $n = 102$ mice). The $Tfam$ genotype and cre recombinase gene were identified at weaning by PCR analysis of the ear notch biopsy DNA[49].

All animal procedures were in accordance with the UK Animals (Scientific Procedures) Act 1986 and ARRIVE guidelines. Study approval was given by Comparative Biology Centre (CBC) at Newcastle University, UK. Mice were housed in groups of 2–6 mice in a ventilated cage and placed on the normal 12-h light-dark cycle with lights on at 07:00 am with food and water provided *ad libitum*. Experiments were conducted on both male and female mice. To ensure the welfare of animals was not compromised, a daily clinical scoring system was utilised to monitor symptom emergence and progression from 5 weeks of age (Supplementary Table 1).

**Behavioural testing.** Numerous behavioural tests of the mice were performed. (1) Accelerating rotating rod, or rotarod[50], the assessment was carried out bi-weekly from five weeks of age on the first cohort of mice. Rotarod test was performed using accelerating mode from 5 to 40 RPM in a 5-min period. Three trials were carried out for all mice with a 10-min interval between trials. The mean latency period and maximal speed at which mice can no longer maintain the balance of three trials were recorded. (2) Open-field assessment using Mouse-Trapp device[51] was carried out on the first cohort of mice bi-weekly from six weeks of age, with each mouse paw touch detected via touch Samsung tablet as previously described. At 8–9 weeks of age, mice were additionally tested using (3) visual cliff assessment to investigate visual depth perception[52] (second cohort of mice); (4) elevated plus maze (EPM) to examine anxiety levels[53] (second cohort of mice); (5) novel object recognition (NOR) test to assess working and recognition memory[54] (first cohort of mice). During the EPM test, mice were placed in the central quadrant and allowed to explore the maze for 10 min. If all four paws of the mice were placed either in the open or closed arm of the maze it was counted as an entry. In the NOR test, briefly, mice underwent a 3-min acquisition phase during which they were presented with two identical objects within the arena, followed by the inter-trial interval of 1 min (retention phase) where the mouse was placed in the holding cage, before being presented with one familiar and one novel object in the test phase for 3 min. Objects, as well as their respective positions in the arena, were alternated for each test to avoid any confounding factors such as any innate preferences for objects themselves or their placement within the arena[55]. The discrimination index between novel and familiar objects was calculated by dividing the difference in time spent exploring novel and familiar objects by the total time spent exploring both objects during the test phase. Moreover, the number of stargazing episodes was quantified in the knockout and littermate control mice of 10 weeks of age using video recordings of 10 min in length blinded to the genotype of the animal and by two independent assessors. The number of

stargazing episodes was averaged between the assessors and calculated per minute.

**Isolation of individual Purkinje cells and mtDNA copy number analysis.** Animals were sacrificed by anaesthesia using inhalational isoflurane, followed by intramuscular injection of ketamine ($\geq 100$ mg kg$^{-1}$) and xylazine ($\geq 10$ mg kg$^{-1}$) and transcardially perfused prior to brain dissection as previously described[56]. Subdissected cerebellum was snap-frozen in cooled isopentane, and samples were stored in $-80$ °C. The cerebellum was cryostat-sectioned in coronal plane at a thickness of 15 μm. Air dried sections was subjected to standard haematoxylin and eosin staining. Sections were rapidly dehydrated in ascending ethanol gradient (70, 95, 100%) and left to air dry. At least fifty pooled Purkinje neurons per mouse were laser-microdissected using the PALM MicroBeam (Zeiss). Neurons were captured in 10 μl of lysis buffer (0.5 M Tris-HCL pH 8.5, 0.5% tween 20, 1% proteinase K). Samples were then centrifuged, followed by lysis (55 °C for 3 h, 95 °C for 10 min for Proteinase K inactivation) and stored at $-20$ °C. To determine mtDNA copy number, quantitative real-time PCR was performed on a CFX96 Touch Real-Time PCR Detection System (Bio-Rad) using single plex Taqman assay targeting mouse $MT$-$ND5$ gene (Supplementary Table 2) as previously described[57] with forward and reverse primers: 5′-ACCTAATTAAACACATCAACTTCCC-3′ and 5′-GACT-CAGTGCCAGGTTGTAA-3′, respectively. Briefly, this involved initial denaturation at 95 °C for 3 min followed by 40 cycles of denaturation for 10 s, annealing and extension at 58 °C for 1 min. Standard curves using PCR-generated templates were used for absolute quantification. Samples, including no-template and lysis buffer-only controls and standards were measured in triplicate. Copy number data were normalised to the cumulative surface area of the pooled of Purkinje neurons isolated from each section.

**Immunohistochemistry.** Animals were anaesthetised as described above. All main reflexes (eye blinking, pedal withdrawal, righting reflex and tail withdrawal) were checked to ensure a lack of neurological response prior to transcardial perfusion with cold saline solution, followed by cold 4% PFA in 0.1 M PBS. Brains were removed and post-fixed in formalin overnight, then transferred to 70% ethanol prior to processing and paraffin embedding. Formalin-fixed paraffin-embedded (FFPE) blocks were then cut on a microtome to generate 5 μm-thick sections mounted on SuperFrost slides. Sections were deparaffinised in a 60 °C oven for at least 20 min, followed by immersion in Histo-Clear (National Diagnostics) and rehydrated through graded ethanol series (100, 100, 95, 70%) to distilled water. Antigen retrieval was conducted by using microwave-boiled 10 mM sodium citrate at pH 6.0 for 10 min or with 1 mM EDTA at pH 8.0 buffer using the 2100 retriever unit (Electron Microscopy Sciences) for 40 min. Sections were washed in distilled water prior to incubation with 3% (v/v) $H_2O_2$ to block endogenous peroxidase activity. Subsequently, sections were washed with TBS with 0.05% (v/v) Tween 20 (TBST) buffer prior to incubation with 10% (v/v) normal goat serum in TBS for 1 h at room temperature. Avidin-biotin blocking stage was carried out according to manufacturer's instructions (SP-2001, Vector Laboratories, UK). Primary antibodies against protein targets were applied overnight at 4 °C (Supplementary Table 3). Sections were washed in TBST, and then secondary biotinylated anti-mouse or anti-rabbit antibodies were added at dilution 1:200 in 10% normal goat serum made up in TBST the next day for 2 h at 4 °C. Following further TBST washes, the horseradish peroxidase reaction was carried out for 30 min at room temperature using Vectastain Elite ABC HRP kit (Vector Laboratories, UK). Sections were finally washed in TBST

and water prior to the addition of DAB chromogen substrate in DAB stable buffer for 5 min at room temperature. Sections were thoroughly washed in water, dehydrated through graded ethanol series, and bathed in Histo-Clear prior to cover-slipping the slides with DPX.

**Microscopy and 2D cell counting.** Formalin-fixed paraffin-embedded (FFPE) sections were examined under a stereology light microscope Olympus BX51 (Olympus Corporation), equipped with motorised specimen stage and 4x–100x Apo objectives and CCD colour video camera and Stereo Investigator software (MBF Bioscience). A freehand outline of the region of interest (ROI) was drawn at 4x magnification and a meander scan was performed at 40x with cells positive for DAB product counted. Cell counts were normalised to the surface area to generate cell density or per unit length of the Purkinje cell layer to generate normalised Purkinje neuron numbers. Purkinje neurons were counted only within the areas where Purkinje cells were arranged in a single lamina.

**Immunofluorescence.** All primary and secondary antibodies used for immunofluorescence (IF) are described in Supplementary Tables 4 and 5. FFPE sections were deparaffinized and rehydrated, followed by antigen retrieval as described above. Sections were then blocked in 10% normal goat serum, and depending on the protocol, followed by avidin and biotin blocking steps. Primary antibodies were applied overnight at 4 °C. One section was included as a no-primary antibody control. Sections were washed in TBST prior to the application of Alexa Fluor-conjugated secondary antibodies (Life Technologies) for 2 h at 4 °C at 1:100 dilution. Anti-rabbit biotinylated antibody (1:200) was used to enhance PV signal, if necessary, followed by an incubation with a secondary antibody cocktail consisting of streptavidin-conjugated Alexa Fluor 405. Quadruple immuno-fluorescence experiment in human FFPE Brodmann area 17 (BA17) occipital lobe was performed using EDTA pH 8.0 antigen retrieval and primary antibodies against PV (1:1500 dilution, Swant PV27), NDUFB8 (1:100 dilution, Abcam ab110242), COXI (1:200 dilution, Abcam ab14705) and porin (1:200 dilution, Abcam ab14734). Secondary antibodies used were goat anti-rabbit IgG Alexa Fluor-405, anti-mouse IgG2a AlexaFluor-488, anti-mouse IgG1 Alexa Fluor-546 and anti-mouse IgG2b Alexa-Fluor-647. For all staining procedures, sections were incubated with 3% Sudan Black in 70% ethanol for 10 min to quench the autofluorescence. Sections were washed in distilled water and mounted with ProLong Gold (Life Technologies).

**Confocal microscopy and semi-quantitative analysis of protein expression.** Neurons were imaged in each ROI on $x$ and $y$-planes using a point scanning microscope (Nikon A1+) with either a 20x objective with a numerical aperture of 0.75 or an oil-immersion 60x objective with a numerical aperture of 1.4 using NIS Elements software for image acquisition. Four lasers were used (405, 488, 561 and 647 nm), each detecting a signal from a different fluorophore in channel series mode. Nikon A1+ ND2 Images were imported to FIJI software for measurement of signal intensities in the cells of interest using a freehand drawing tool. Alternatively, an LSM-880 (Carl Zeiss) microscope was used with Airyscan detection super-resolution mode equipped with 355, 488, 561, 633 nm lasers and an oil-immersion 63x objective with numerical aperture of 1.4 using ZEN Black 2.3 SP1 software for image acquisition and ZEN Blue or ZEN Lite software for image analysis. Mitochondrial content (NDUFA13; 488 nm channel and SDHA; 633 nm channel) within the inhibitory axonal terminals surrounding deep cerebellar nuclei (DCN) neurons was measured

in GAD1-2-positive (355 nm channel) "objects" using the oil-immersion 63x objective lens at 4x scanning zoom, image size of $1064 \times 1064$ pixels and 0.18 μm step size for $z$-stacks. Z-stack images were acquired to image microglia (identified via Iba-1 signal in the 355 nm channel) using an oil-immersion 63x objective lens at optimal settings of 0.16 μm step size and $848 \times 848$ pixels image size at 5x scanning zoom. All confocal settings, including laser intensity, gain, offset, and zoom were held constant for a given channel.

2D images were analysed using Fiji, ZEN Blue or ZEN Lite softwares and z-stacks were analysed using Volocity software. Z-stacks were taken on the Zeiss LSM-880 confocal microscope with Airyscan processing and analysed in 3D using Volocity in "quantitation" mode. The channel corresponding to the cell marker of interest (e.g., either GAD1-2 or Iba-1) thresholds were set to detect inhibitory axons and their terminals or microglia within each stack (ROI) and mean signal intensity for each channel and volume were recorded for each ROI.

The density values for each protein target were quantified and log transformed to normalise data. Mitochondrial proteins were normalised against either succinate dehydrogenase subunit A (SDHA; complex II) or porin that were used as markers of mitochondrial mass. The mean and SD values were calculated for each mitochondrial protein/mitochondrial mass marker ratio in control cells and these parameters were used to calculate $z$-scores for all immunofluorescence data. Cells were classified based on SD limits. For proteins NDUFB8[12], NDUFA13[11], SDHA, UqCRC2, COXI[12], COXIV, ATP-B, porin, pyruvate carboxylase (PC), GAD1-2 and tyrosine hydroxylase (TH): overexpression if $z > 2$, normal if $-2 < z < 2$ SD, low if $z < -2$ SD, deficient if $z < -3$ SD and severely deficient if $z < -4$ SD.

**Clinical assessment of patients.** In total, 11 adult patients with genetically and clinically confirmed mitochondrial disease were included in this study. All patients had a well-characterised clinical phenotype (Table 1). Patients harboured either mtDNA pathogenic point variants, such as m.3243 A > G ($n = 5$), m. 8344 A > G ($n = 3$), or biallelic pathogenic variants in *POLG* ($n = 3$). Patients were clinically assessed using a validated rating scale termed Newcastle Mitochondrial Disease Adult Scale (NMDAS)[58], neuroimaging and electroencephalogram, where available, and evidence of neurological impairments that included stroke-like episodes, ataxia, epilepsy, and cognitive impairment (Table 1).

**Post-mortem brain tissue.** *Post-mortem* brains were processed as previously described[59]. Our neuropathological investigation focused on the occipital lobe, specifically the primary visual cortex (Brodmann area 17; BA17), as this is the area highly susceptible to seizures and neuropathological changes in mitochondrial disease[6]. 16 age-matched control subjects were used with no history of seizures or strokes (Supplementary Table 6). The mean age of the patient group was 52.6 years (95% CI: 42.8–62.3) and the control group was 55.9 years (95% CI: 49.1–62.7). The sex distribution (female to male) was 2 to 9 in mitochondrial disease and 5 to 11 in controls. There were no statistically significant differences in median age ($P = 0.4286$, Mann–Whitney test), nor in the sex distribution ($P = 0.66$, Fisher's exact test) between the groups. Tissues were obtained from the Newcastle Brain Tissue Resource (NBTR) with ethical approval from Newcastle and North Tyneside 1 REC (Ref 19/NE/0008) as well as Edinburgh Brain Bank, which has ethical approval from the East of Scotland Research Ethics Service REC1.

**Statistics and reproducibility.** Analyses were performed using GraphPad Prism (GraphPad Software, Inc.). Sample sizes

constituted $n \geq 3$ biological replicates per group. All data were tested for normality using Shapiro–Wilk test. Normally distributed data were subjected to either a two-tailed unpaired Student's $t$-test when comparing two groups or one-way ANOVA with Tukey's post hoc test when more than two groups were compared. For non-parametric data, a two-tailed Mann–Whitney $U$-test was used for two groups, and Kruskal–Wallis one-way ANOVA on ranks was used for more than two groups with Dunn's pairwise post hoc comparison. Where required, $t$-tests were performed with Benjamini–Hochberg adjustment to control for false discovery rate. To compare proportion levels between the two groups, a two-tailed Fisher's exact or chi-squared test was used.

The nlme package within R statistical software (R Core Team 2022) was used to perform linear mixed-effects models[60], allowing comparisons to be made between groups whilst accounting for variation within each individual mouse or case. Within each model, the group was included as a fixed effect and the mouse or case as a random effect, with residuals checked for normality. The level of $P$-value significance was set to 0.05 (*) with further classification in the degree of significance: $P < 0.01$ (**), $P < 0.001$ (***), $P < 0.0001$ (****). Presented graphs contain all data points as boxplots with inter-quartile ranges (IQR) or have mean ± SD indicated, unless otherwise stated.

**Reporting summary**. Further information on research design is available in the Nature Portfolio Reporting Summary linked to this article.

## Data availability

All source data for figures is provided in the Supplementary Data file.

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

## Acknowledgements

This work was supported by the Wellcome Centre for Mitochondrial Research (203105/Z/16/Z) and EAO was additionally supported by the Newcastle University Overseas Research Scholarship and Academic Development Post-Submission Scholarship. Tissue for this study was provided by the Newcastle Brain Tissue Resource, which is funded in part by a grant from the UK Medical Research Council (G0400074), by NIHR Newcastle Biomedical Research Centre, and as part of the Brains for Dementia Research Programme jointly funded by Alzheimer's Research UK and Alzheimer's Society. Tissue samples were also obtained from the Edinburgh Brain and Tissue Bank which is funded by the Medical Research Council. We thank the staff of the Centre for Comparative Biology, Faculty of Medical Sciences, Newcastle University, for their help with animal husbandry and Bioimaging Facility, Faculty of Medical Sciences, Newcastle University, for their help with confocal microscopy.

## Author contributions

E.A.O., C.B., N.Z.L. conducted experiments and acquired data for the study. A.B., E.A.O., D.A., N.Z.L. performed data analyses and E.A.O. and A.B. performed statistical analyses. E.A.O., C.B., A.B., N.Z.L., D.M.T., F.L.B., Y.S.N., G.S.G. contributed to the design of the study and interpretation of data. All authors have critically revised the manuscript and given their final approval.

## Competing interests

The authors declare no competing interests.

## Ethics approval

This research included local researchers throughout the research process, including study design, study implementation, data ownership, intellectual property, and authorship of publications. The research is locally relevant. The roles and responsibilities were agreed among collaborators ahead of the research and capacity-building plans for local researchers were discussed. The study has been approved by a local ethics review committee. The research does not result in stigmatisation, incrimination, discrimination, or otherwise personal risk to participants. The research was conducted in line with health and safety regulations at Newcastle University, where all appropriate guidelines on chemical and biological safety, as well as relevant risk assessments for all procedures, were followed. We have taken local and regional research relevant to your study into account in citations, which was conducted at Newcastle University.
