## [Peer Review File · Communications Biology]

Reviewers' comments:

Reviewer #1 (Remarks to the Author):

Olkhova et al. report a novel mouse model harboring conditional knockout of Tfam in parvalbumin-expressing neurons. This is a very timely study, given the recent and increasing evidence for selective vulnerability of GABAergic neurons to OXPHOS dysfunction. The work is robustly performed, and the paper is well-written, although it does contain a few typos. The mouse itself is an important addition to the growing arsenal of neurological models of mitochondrial dysfunction, and I expect that it will enable significant basic, translational, and even preclinical research.

I recommend this work for publication, subject to relatively minor revisions. Please see my comments and suggestions below:

1) Abstract, line 10: «patients with mtDNA and POLG variants». Please change to "mutations", as not all variants are pathogenic.

2) Introduction, lines 4-6: "The central nervous system is frequently affected in patients with mitochondrial disease, manifesting with cerebellar ataxia, stroke-like episodes, epilepsy, extrapyramidal movement disorders, and cognitive impairment" – I suggest changing to "...commonly manifesting with...". because there are many other neurological manifestations (e.g., peripheral neuropathy, headache, etc).

3) Introduction, lines 8-9: "To develop novel treatments for these conditions, it is crucial to delineate the mechanisms contributing to debilitating symptoms such as ataxia and epilepsy in mitochondrial disease." – redundant to say both "for these conditions" and "in mitochondrial disease" – please rephrase

4) Page 6 line 4: The authors state that the model can be used to "mechanistically explore the hypothesis of selective PV+ cell vulnerability to mitochondrial dysfunction". Since PV+ cells are selectively and specifically targeted in this model, I do not see how it can be used to assess the selective PV+ cell vulnerability to mitochondrial dysfunction. This would require a generalized mitochondrial defect affecting the entire nervous system – or at least all neurons, so that the vulnerability of PV+ positive cells can be compared to that of other neuronal types. I would rather say that this model allows us to study the events and processes downstream to mtDNA depletion in PV+ positive cells. I believe that this statement should be rephrased,

5) Supplementary Fig. 2: I assume that each point represents the mean or median of several PCs tested per mouse? This should be stated. "Graph indicates mean \pm SD.", please consider changing to "Bars indicate mean \pm SD".

Please check whether a similar clarification should be provided with other figure as well.

6) There are some mentions of "unpaired Student's t-test" – it is not necessary to specify "unpaired" when there is clearly no paired data being analyzed.

7) Page 13 lines 4-6: "Taken together, these findings demonstrate that the glial activation within cerebellum is a result of mitochondrial dysfunction selectively in PV+ interneurons and/or loss of Purkinje neurons, without any primary genetic mitochondrial impairments within the glial cells." I believe this statement is unnecessarily complicated and perhaps redundant. Glial activation (astro- and microgliosis) is a rather unspecific response accompanying neuronal damage and death due to a multitude of acute, subacute and chronic etiologies. Given that PCs are affected in the model, it comes

to no surprise that we observe significant gliosis. I believe this argument should be downplayed or removed.

8) Page 13 lines 17-19: We identified that in control tissues, expression of porin, complex I subunit encoded NADH:ubiquinone oxidoreductase 8 (NDUFB8) and complex IV subunit COXI was upregulated in PV+ interneurons in relation to non-PV-expressing cells.

(a) Is there a typo in "complex I subunit encoded NADH:ubiquinone oxidoreductase 8 (NDUFB8)"? I believe NDUFB8 stands for "NADH:ubiquinone oxidoreductase subunit B8"?

(b) I would not use the term "upregulated" here, but simply say that the expression was "higher". The terms "up-/downregulated" imply a change/regulation is taking place (i.e., something increases, or decreases from its original state). This is not the case when comparing different cell types, however.

9) Page 13 lines 22-24: "Differentially affected complex I and IV expression in 22 our mitochondrial disease patient cohort is consistent with previously published data in Purkinje neurons of the cerebellum and inhibitory interneurons in the cerebral cortex in patients with mitochondrial disease." Consider saying "...in patients with these mitochondrial diseases.". Generalizing would be incorrect as there are mitochondrial diseases that affect mainly CIV.

10) Page 13 lines 25-27: "Moreover, an upregulation of mitochondrial mass marker porin was observed in PV+ interneurons in patients with mitochondrial disease (Fig. 10c; $P < 0.0001$, Mann-Whitney test), which further validates mouse model data in the cerebral cortex"
I wouldn't call this a validation, but rather a corroboration/support.

11) Unless this is dictated by the journal it is always preferable to give exact p-values rather than "<" or ">" – you can use power of 10 if the value is too small.

Reviewer #2 (Remarks to the Author):

Overall, there is tremendous enthusiasm for how this study of TFAM KO in PV neurons is considerably more comprehensive in its analyses of various PV-expressing subgroups (including Purkinje Cells and TRN) rather than the more typical focus on cortical interneurons only.

However, there are many important weaknesses, particularly in the statistics, detailed below.

Fig. 2. Since OXPHOS itself is not being measured, the fig title is not accurate. Levels of OXPHOS-functioning proteins are imaged.

Fig. 4.

Specificity of PGC1a immuno has traditionally been challenging. Do the authors expects PGC1A signal to overlap with VDAC? If not, why not, and if so, can they add a supplement to show this? Any other evidence as to the specificity of this AB—for example does it lack signal in PGC1a KOs?

The statistics seem to be a Mann Whitney U on all 100+ neurons counted from each group. This is meaningful but would mask a major skewing produced by one animal whose many counts are different from the others of the same group, and in general the MWU entails the assumption that individual data points are from reasonably independent entities (like comparisons of people's heights between countries not including more than one from the same family). What happens if they do a more rigorous test and first check for normality, then average the results for each individual mouse and run

a T-test with N=4 per group?

This issue is more concerning in Fig. 5, where sometimes an N of 3 mice per group is evaluated via N=hundreds of count of neurons from the same animal, and there is no reporting of the average/Standard Deviation of each animal per group.

For example "e hippocampus (P = 0.001, Student's t-test; n = 15 neurons from 2 mice in control group and n = 17 neurons from 4 mice in knockout group)." It is not appropriate to consider a data set to be statistically significant by comparing many measures of two individuals in one group to 4 individuals of another using N of total neurons counted per group. That said, from this reviewer's perspective, it would be appropriate to include the data, perhaps in the supp, with a table or color coded dot plot to indicate how the average values/SD of neurons from the same individuals of each group compare to each other within group and to the other group. Even though not "significant" in that with an N=2 it is not possible to know what the "real" population being sampled looks like, how the trend implied from the hippocampal data fits with other regions evaluated in more detail is interesting.

Supp. Fig. 3. Again, would a t-test, adjusted for multiple comparisons (4 regions tested), using N=4 animals per group, be significant?

Fig. 9—mention pt. data in fig title.
Correction for multiple comparisons?

Of note, when one sees the same direction of change across multiple measures that could reasonably be considered associated, for example complex I deficiency in the KOs across multiple tissues, a correction for multiple comparison's simply punishes the experimenter for testing multiple regions. That said, in several places a false discovery rate is warranted.

In the discussion, it would be useful to comment on the interesting disparity across PV cells in their capacity or approach to compensate for the tfam loss.

Reviewer #3 (Remarks to the Author):

In this paper, Olkhova et al, the authors report the generation of a new mouse model with a conditional knockout of nuclear-encoded Tfam selectively in PV+ neurons. They studied the phenotype of this model to understand the neuropathological effects of mitochondrial dysfunction in PV+ cells in vivo.

This is a nicely and comprehensively conducted and described study. It is also informative. The statistical methods were appropriate and sufficiently robust to answer the stated research question. I have a few comments.

- 1- The two subsections: "mtDNA depletion in Purkinje cells" and "Brain region-dependent mitochondrial mass alterations and OXPHOS deficiencies in PV+ neurons in knockout mice" should be in opposite order.
- 2- It is not clear whether the same animals were used for all behavioral experiments and what is the order?
- 3- The number of animals in some behavioral experiments seem low to ensure statistical power, such as Fig. 1b, 1c and 1d
- 4- In other behavioral experiments, there are huge differences in the number of animals between the control and the experimental groups

5- In the immunohistochemistry experiments, how did authors ensure all sections have similar background. Intensity measurement in immunohistochemistry experiments is not very accurate quantitative method, I wonder if there is a better method that authors could use.

Reviewers' comments:

Reviewer #1 (Remarks to the Author):

Olkhova et al. report a novel mouse model harboring conditional knockout of Tfam in parvalbumin-expressing neurons. This is a very timely study, given the recent and increasing evidence for selective vulnerability of GABAergic neurons to OXPHOS dysfunction. The work is robustly performed, and the paper is well-written, although it does contain a few typos. The mouse itself is an important addition to the growing arsenal of neurological models of mitochondrial dysfunction, and I expect that it will enable significant basic, translational, and even preclinical research.

I recommend this work for publication, subject to relatively minor revisions. Please see my comments and suggestions below:

We sincerely thank the reviewer for recognising the importance of our novel mouse model and its implications for the research field. We have incorporated all the proposed changes as suggested, as outlined below.

1) Abstract, line 10: «patients with mtDNA and POLG variants». Please change to “mutations”, as not all variants are pathogenic.

We have added the adjective “pathogenic” before the word “variants” to denote this.

2) Introduction, lines 4-6: “The central nervous system is frequently affected in patients with mitochondrial disease, manifesting with cerebellar ataxia, stroke-like episodes, epilepsy, extrapyramidal movement disorders, and cognitive impairment” – I suggest changing to “...commonly manifesting with...”. because there are many other neurological manifestations (e.g., peripheral neuropathy, headache, etc).

We have amended exactly as requested (page 4, line 5).

3) Introduction, lines 8-9: “To develop novel treatments for these conditions, it is crucial to delineate the mechanisms contributing to debilitating symptoms such as ataxia and epilepsy in mitochondrial disease.” – redundant to say both “for these conditions” and “in mitochondrial disease” – please rephrase

We have amended as requested (page 4, lines 8-9) and deleted “for these conditions” in the sentence.

4) Page 6 line 4: The authors state that the model can be used to “mechanistically explore the hypothesis of selective PV+ cell vulnerability to mitochondrial dysfunction”. Since PV+ cells are selectively and specifically targeted in this model, I do not see how it can be used to assess the selective PV+ cell vulnerability to mitochondrial dysfunction. This would require a generalized mitochondrial defect affecting the entire nervous system – or at least all neurons, so that the vulnerability of PV+ positive cells can be compared to that of other neuronal types.

I would rather say that this model allows us to study the events and processes downstream to mtDNA depletion in PV+ positive cells. I believe that this statement should be rephrased,

We thank reviewer for this insightful comment and have rephrased it as follows on page 6 lines 4 – 7:

We mechanistically explored the hypothesis of whether selective PV⁺ cell mitochondrial dysfunction would be sufficient to induce a neurological phenotype reminiscent of human mitochondrial disease¹⁴ by creating a suitable in vivo mitochondrial disease model based on available neuropathological data¹⁵. This model would allow us to study the events and processes downstream of mtDNA depletion in PV⁺ cells.

5) Supplementary Fig. 2: I assume that each point represents the mean or median of several PCs tested per mouse? This should be stated. "Graph indicates mean \pm SD.", please consider changing to "Bars indicate mean \pm SD".

Please check whether a similar clarification should be provided with other figure as well.

Many thanks for this important point. We have revised the Figure legend for **Supplementary Figure 2**, specifying that 50 Purkinje cells were pooled for each mouse, and we added the information on mean \pm SD indicated by bars. Please see below the amended figure legend:

Supplementary Fig. 2: mtDNA depletion in pooled Purkinje neurons per mouse. Mean mitochondrial DNA copy number was significantly lower in Purkinje neurons in the knockout group in comparison to littermate control animals ($P = 0.0132$, Student's t -test; $n = 4$ mice per group). Each point indicates pooled mtDNA copy number per mouse per mm² of laser capture microdissection area that was occupied by 50 Purkinje neurons. Graph indicates mean \pm SD.

6) There are some mentions of "unpaired Student's t-test" – it is not necessary to specify "unpaired" when there is clearly no paired data being analyzed.

We have amended this as requested throughout the manuscript. The word "unpaired" has been removed as rightfully highlighted by the reviewer, in that there were no paired analyses carried out in this programme of work.

7) Page 13 lines 4-6: "Taken together, these findings demonstrate that the glial activation within cerebellum is a result of mitochondrial dysfunction selectively in PV⁺ interneurons and/or loss of Purkinje neurons, without any primary genetic mitochondrial impairments within the glial cells."

I believe this statement is unnecessarily complicated and perhaps redundant. Glial activation (astro- and microgliosis) is a rather unspecific response accompanying neuronal damage and death due to a multitude of acute, subacute, and chronic etiologies. Given that PCs are affected in the model, it comes to no surprise that we observe significant gliosis. I believe this argument should be downplayed or removed.

We have carefully considered this extremely constructive feedback, and we have now removed this sentence as advised from the revised manuscript.

8) Page 13 lines 17-19: We identified that in control tissues, expression of porin, complex I subunit encoded NADH:ubiquinone oxidoreductase 8 (NDUFB8) and complex IV subunit COXI was upregulated in PV⁺ interneurons in relation to non-PV-expressing cells.

(a) Is there a typo in “complex I subunit encoded NADH:ubiquinone oxidoreductase 8 (NDUFB8)”? I believe NDUFB8 stands for “NADH:ubiquinone oxidoreductase subunit B8”?

Thank you to the reviewer for this comment. This is indeed a typographical error which we have corrected in this revised version of the manuscript.

(b) I would not use the term “upregulated” here, but simply say that the expression was “higher”. The terms “up-/downregulated” imply a change/regulation is taking place (i.e., something increases, or decreases from its original state). This is not the case when comparing different cell types, however.

We agree with the reviewer and have removed the terms “up/downregulated” throughout the manuscript.

9) Page 13 lines 22-24: “Differentially affected complex I and IV expression in 22 our mitochondrial disease patient cohort is consistent with previously published data in Purkinje neurons of the cerebellum and inhibitory interneurons in the cerebral cortex in patients with mitochondrial disease.”

Consider saying “...in patients with these mitochondrial diseases.”. Generalizing would be incorrect as there are mitochondrial diseases that affect mainly CIV.

We thank the reviewer for this helpful comment and have revised the sentence as follows (please note it is now on page 15 lines 26 – page 16 line 1 in the revised manuscript):

Specifically, we report PV+ Purkinje neuron loss, with surviving Purkinje neurons exhibiting combined complex I and IV severe deficiencies, resembling neuropathological findings reported in human post-mortem tissues from adult and paediatric patients with these mitochondrial diseases^{9,11,13}.

10) Page 13 lines 25-27: “Moreover, an upregulation of mitochondrial mass marker porin was observed in PV+ interneurons in patients with mitochondrial disease (Fig. 10c; $P < 0.0001$, Mann-Whitney test), which further validates mouse model data in the cerebral cortex”

I wouldn't call this a validation, but rather a corroboration/support.

Thank you for highlighting this. This sentence is now redundant following extensive statistical re-analysis (as advised by Reviewer 2) and has been removed.

11) Unless this is dictated by the journal it is always preferable to give exact p-values rather than “<” or “>” – you can use power of 10 if the value is too small.

We thank you for this observation. We have amended as requested – all figure legends now provide exact P values for each test used up to four decimal places.

Reviewer #2 (Remarks to the Author):

Overall, there is tremendous enthusiasm for how this study of TFAM KO in PV neurons is considerably more comprehensive in its analyses of various PV-expressing subgroups (including Purkinje Cells and TRN) rather than the more typical focus on cortical interneurons only.

However, there are many important weaknesses, particularly in the statistics, detailed below.

We thank the reviewer for their enthusiasm of our work and highlighting its novelty. We are most appreciative of the highly constructive feedback with regards to statistical analyses and have revised as suggested. Please see comments that follow as to how we have addressed each point raised, in the revised manuscript.

Fig. 2. Since OXPHOS itself is not being measured, the fig title is not accurate. Levels of OXPHOS-functioning proteins are imaged.

Thank you for this extremely helpful comment. We have revised the title of Figure 2 as follows:

*Figure 2. **OXPHOS-functioning protein deficiencies in PV+ neurons in PV^{cre}Tfam^{-/-} mice are brain region-dependent.***

Fig. 4. Specificity of PGC1a immuno has traditionally been challenging. Do the authors expects PGC1A signal to overlap with VDAC? If not, why not, and if so, can they add a supplement to show this? Any other evidence as to the specificity of this AB—for example does it lack signal in PGC1a KOs?

Thank you to the reviewer for this comment. In direct response to the query, we do not expect PGC1 to fully colocalise with VDAC1, as PGC1 is a transcriptional factor and will also be distributed to the nucleus. However, there may be some overlap with VDAC1 (as demonstrated on merged images from original manuscript Figure 4 and **revised Supplementary Figure 7**).

To address the latter point and to our best knowledge, we are not aware of any studies that have validated this antibody in PGC1-knockout cells.

The statistics seem to be a Mann Whitney U on all 100+ neurons counted from each group. This is meaningful but would mask a major skewing produced by one animal whose many counts are different from the others of the same group, and in general the MWU entails the assumption that individual data points are from reasonably independent entities (like comparisons of people's heights between countries not including more than one from the same family). What happens if they do a more rigorous test and first check for normality, then average the results for each individual mouse and run a T-test with N=4 per group?

We thank reviewer for this extremely helpful comment.

We have now extensively revised the statistical analysis of the data to address the points raised. We have sought advice from our biostatistician (Alasdair Blain). **To account for the potentially large influence of the neurons of one mouse on a result analysed at group level, we have now performed linear mixed-effect models in place of Mann Whitney U tests on single cells.** Mixed effect models allow us to examine the differences between groups whilst considering the variability both within and between individuals in each group, by including mouse as a random effect. This type of model is also less sensitive to unbalanced data than previously used methods. This method allows us to draw conclusions more confidently about the variation between groups.

Following the reanalysis of the PGC1 and porin data using linear mixed-effects model from the original Figure 4, it was found to not be statistically significant, and as a result

we have now moved this Figure to the Supplementary Figures (please kindly note that the original Figure 4 is now **Supplementary Figure 7**).

In the case of *t*-tests performed with averaged data, normality tests were carried out.

This issue is more concerning in Fig. 5, where sometimes an N of 3 mice per group is evaluated via N=hundreds of count of neurons from the same animal, and there is no reporting of the average/Standard Deviation of each animal per group.

For example "e hippocampus ($P = 0.001$, Student's *t*-test; $n = 15$ neurons from 2 mice in control group and $n = 17$ neurons from 4 mice in knockout group)." It is not appropriate to consider a data set to be statistically significant by comparing many measures of two individuals in one group to 4 individuals of another using N of total neurons counted per group. That said, from this reviewer's perspective, it would be appropriate to include the data, perhaps in the supp, with a table or color coded dot plot to indicate how the average values/SD of neurons from the same individuals of each group compare to each other within group and to the other group. Even though not "significant" in that with an $N=2$ it is not possible to know what the "real" population being sampled looks like, how the trend implied from the hippocampal data fits with other regions evaluated in more detail is interesting.

We thank the reviewer for their constructive feedback. To address the statistical query, we have revised the statistical analysis and now performed a **linear mixed-effects model as described above, which accounts for variability in the number of cells analysed per mouse**, to check for mouse-specific effects in pyruvate carboxylase expression, as well as the tyrosine hydroxylase expression in Purkinje neurons for the data in this Figure. To reflect this change, we amended the figure legend (please kindly note this was Figure 5 in the original manuscript; now it is **Figure 4** in the revised manuscript). For your convenience, please see the revised figure legend copied below:

Fig. 4: Metabolic remodeling via pyruvate carboxylase expression, accompanied by signs of hyperexcitability such as ectopic expression of tyrosine hydroxylase and c-Fos expression in Purkinje neurons of the knockout mice.

a Example confocal micrographs of Purkinje neurons (green) and pyruvate carboxylase (PC) co-staining (red) in littermate control (top) and knockout mouse cerebellum (bottom). Scale bars – 10 μ m. **b** Increase in the percentage of Purkinje neurons displaying overexpression of PC. $P = 0.0681$, linear mixed-effects model; $n = 107$ neurons from 3 control and $n = 230$ from 6 knockout mice. **c** Increase in the percentage of PV⁺ neurons of the TRN with PC overexpression in the knockout mice. $P = 0.5743$, linear mixed-effects model; $n = 54$ neurons from 2 control and $n = 108$ neurons from 3 knockout mice. **d** Example confocal micrographs of Purkinje neurons expressing tyrosine hydroxylase. **e** Quantification of optical density z-

scores of tyrosine hydroxylase in individual Purkinje neurons revealed a non-significant increase in the knockout mouse group ($P = 0.1022$, linear mixed-effects model; $n = 166$ neurons from 5 control and $n = 228$ from 6 knockout mice). **f** Example light micrograph of c-Fos immunohistochemical staining in Purkinje neurons in the knockout cerebellum. Arrowheads show neurons with positive c-Fos signal detected within nuclei. Scale bar – 100 μm . **g** Graph demonstrates the number of c-Fos-immunoreactive Purkinje cells per unit length of the Purkinje cell layer ($n = 4$ control and $n = 6$ knockout mice). **h** c-Fos-expressing neuronal density in the DCN was non-significantly increased in the knockout group ($P = 0.1667$, Mann-Whitney test; $n = 3$ control and $n = 6$ knockout mice).

Having performed extensive statistical re-analysis of the data sets, we now note the lack of statistical significance, as the revised P -values for pyruvate carboxylase changes in Purkinje and TRN neurons are non-significant. Similarly, tyrosine hydroxylase expression P -value indicates a lack of significance. We also moved the cortical and hippocampal pyruvate carboxylase data, which were non-significant, to the supplementary data to focus readers' attention in the revised Figure 4 on Purkinje neurons mainly (these data have now been moved to the Supplementary Figure 8).

To considerably enhance the manuscript, we have implemented changes to data analysis as suggested by the Reviewer 2 to other data as well. For all the other graphs showing individual cell data points across the entire manuscript, we amended these from displaying individual cell data points to averaged data per mouse or per case and performed linear mixed-effects model or t -tests or Mann-Whitney U tests depending on the outcome of Shapiro-Wilk normality test, such as in **Figures 6, 7, 8, 9c**.

All data shown **Figures 6, 7, 8** remain statistically significant (except Figure 7c), despite averaging all data per mouse or per case. **Figure 9c** P -value for porin (averaged data per case) was no longer significant. We thank the reviewer for these comments and feel that, while the key original findings of our manuscript remain, we have now greatly enhanced the quality of the manuscript because of the reviewer's feedback and resultant suggested changes.

Supp. Fig. 3. Again, would a t -test, adjusted for multiple comparisons (4 regions tested), using $N=4$ animals per group, be significant?

We have now performed a linear mixed-effects model to check for mouse-specific effects for porin in **Supplementary Figure 3** and the same analyses for NDUFA13, COXIV across all the brain regions analysed in **Supplementary Figure 4**. We have amended these graphs and P -values accordingly. We have also performed the same analysis for SDHA expression in **Supplementary Figure 5**, which demonstrated no significant changes between genotype groups.

Following data averaging per mouse and linear mixed-effects model analyses of data from various brain regions, we find that **decreases in complex I and complex IV**

expression remain significant in Purkinje neurons, TRN neurons, PV⁺ neurons of the molecular layer of the cerebellum and posterior cerebral cortical regions (Supplementary Figure 4). To boost the sample size for the cortical region which suffered from low sample size in the control group, we repeated the experiment to measure NDUFA13, COXIV and SDHA subunit expression in PV⁺ neurons in the posterior cortical regions. The only regions that did not reach statistical significance for complex I and IV subunit expression between genotypes were the deep cerebellar nuclei (DCN) and hippocampus. We have now added this information to the manuscript to enhance clarity and explicitly state this.

Please see the revised Figure Legend for Supplementary Figure 4 below:

Supplementary Fig. 4: NDUFA13 (complex I) and COXIV (complex IV) expression normalised to SDHA is decreased in PV⁺ cells of *PV^{cre}Tfam^{-/-}* mice. **a** Decrease in NDUFA13/SDHA and COXIV/SDHA z-scores per mouse in Purkinje neurons in the knockout animals ($n = 276$ neurons; 6 mice) vs. controls ($n = 130$ neurons from 3 mice) was statistically significant ($P < 0.0001$ and $P = 0.0001$, respectively, linear mixed-effects model). **b** Similarly, in the TRN region, complex I and IV subunits were significantly reduced in the knockout group ($n = 282$ neurons; 6 mice) vs. controls ($n = 119$ neurons; 3 mice) ($P < 0.0001$ and $P = 0.0123$, respectively, linear mixed-effects model), **c** PV⁺ neurons of the molecular layer of the cerebellum demonstrated a significant reduction in complex I and IV subunits ($n = 224$ neurons; 6 mice) vs. controls ($n = 129$ neurons; 4 mice) ($P = 0.0001$ and $P = 0.0014$, respectively, linear mixed-effects model), **d** PV⁺ neurons in posterior cortical areas demonstrated a decrease in complex I and IV subunits in knockout animals (NDUFA13: $n = 65$ neurons from 5 mice; COXIV: 187 neurons from 6 mice) vs. control (NDUFA13: $n = 33$ neurons from 3 mice; COXIV: 97 neurons from 5 mice), which was statistically significant ($P = 0.0066$ and $P = 0.0105$, respectively, linear mixed-effects model). **e** NDUFA13/SDHA expression showed a statistical trend towards reduction in the knockout group in the DCN region ($P = 0.0589$, linear mixed-effects model), whereas COXIV/SDHA expression was unaltered ($P = 0.8716$, linear mixed-effects model; $n = 77$ neurons from 3 control mice and $n = 119$ neurons from 5 knockout mice). **f** NDUFA13/SDHA and COXIV/SDHA expression did not show any differences between the groups in the hippocampus ($P = 0.9361$ and $P = 0.7388$, respectively, linear mixed-effects model; $n = 55$ neurons from 6 knockout mice and $n = 20$ neurons from 3 control mice).

Fig. 9—mention pt. data in fig title.

Correction for multiple comparisons?

In this revised manuscript, **the original Figure 9 is now Figure 8** and we have revised the Figure legend with the Benjamini-Hochberg adjustment to control for the false discovery rate and the data remained significant:

Fig. 8: PV⁺ interneurons of primary visual cortex display a greater overall expression of mitochondrial mass, complex I and complex IV subunits in comparison to non-PV-expressing cells in control tissue and patients with mitochondrial disease frequently demonstrate a decrease in complex I subunit expression.

a Example confocal micrographs demonstrating PV (blue), COXI (green), NDUFB8 (red) and porin (purple) immunofluorescence in primary visual cortex in control and patient 11 (*POLG*). Arrowhead points to non-PV-immunoreactive cell which appears to have reduced mitochondrial mass, NDUFB8 and COXI expression in relation to PV⁺ interneurons in control tissues (top panel). Patient 11 PV⁺ interneuron shows almost complete loss of NDUFB8, decreased COXI, and increased porin signal. Scale bars – 20 μ m. **b-d** Boxplots demonstrating significantly increased porin, NDUFB8 and COXI (not normalised to porin) expression in PV⁺ interneurons vs. non-PV-immunoreactive cells in the occipital lobe of neurologically-normal controls ($P = 0.0235, 0.0131, 0.0131$, respectively; t -test with Benjamini-Hochberg adjustment; $n = 775$ PV⁺ interneurons from 16 controls and $n = 453$ non-PV-immunoreactive cells analysed from 8 controls).

Of note, when one sees the same direction of change across multiple measures that could reasonably be considered associated, for example complex I deficiency in the KOs across multiple tissues, a correction for multiple comparison's simply punishes the experimenter for testing multiple regions. That said, in several places a false discovery rate is warranted.

We thank the reviewer for this comment. We analysed different brain regions to provide clarity on the vulnerability of certain brain regions to mitochondrial dysfunction. We treated each brain region as an independent entity and given the small sample sizes in the biomedical research, we agree with the reviewer that introducing multiple comparisons could result in false negatives, or Type II error. Therefore, we have performed linear mixed-effects model approach instead of t -tests.

In the discussion, it would be useful to comment on the interesting disparity across PV cells in their capacity or approach to compensate for the tfam loss.

We state in the discussion that we speculate that Purkinje neurons are particularly vulnerable to mitochondrial dysfunction due to their high action potential generation rate (which is greater than that of cortical PV+ neurons) and large soma size, which other studies have shown to be linked to a greater ATP demand, for which we provide references.

To ensure clarity, we have now enhanced the sentence in the discussion, please see page 16 of the manuscript lines 1-3 copied below:

We speculate that the disparity in OXPHOS protein defects among PV+ neurons is in part due to Purkinje neurons inherent high rate of firing and tonic mode of spiking⁴⁰ as well as their large cell size⁴¹, which may render this energy-demanding neuronal subtype particularly vulnerable to mitochondrial impairment.

Reviewer #3 (Remarks to the Author):

In this paper, Olkhova et al, the authors report the generation of a new mouse model with a conditional knockout of nuclear-encoded Tfam selectively in PV+ neurons. They studied the phenotype of this model to understand the neuropathological effects of mitochondrial dysfunction in PV+ cells in vivo.

This is a nicely and comprehensively conducted and described study. It is also informative. The statistical methods were appropriate and sufficiently robust to answer the stated research question. I have a few comments.

We are extremely grateful for the complimentary feedback provided by the reviewer.

- 1- The two subsections: “mtDNA depletion in Purkinje cells” and “Brain region-dependent mitochondrial mass alterations and OXPHOS deficiencies in PV+ neurons in knockout mice” should be in opposite order.

Thank you to the Reviewer 3 for your helpful comment, we have made the suggested changes and revised as advised.

- 2- It is not clear whether the same animals were used for all behavioral experiments and what is the order?

We thank you for this important question and for the opportunity to provide clarity.

In some of the experiments it was the same cohort of animals (first cohort) and for other experiments it was a different cohort (second cohort) for the following reasons. This mouse model was characterised in-depth for the first time phenotypically and we had staggered generation of litters which could explain slight differences in sample sizes for the tests. We used the same cohort of animals in the novel object recognition test for cognition, longitudinal open field and rotarod testing which took place biweekly on alternating weeks (rotarod starting at five weeks of age until 12 weeks and open field starting at six weeks of age until ten). Therefore, one cohort of mice was used for NOR, open field, rotarod tests and for animal welfare reasons we could not use the same mice for several additional behavioural tests, as we wished to minimise the number of assessments any mouse had to undergo within a given week.

To add additional behavioural tests to characterise the model in more detail we, therefore, used a separate and smaller cohort of animals (second cohort) to probe for visual depth perception using visual cliff, and elevated-plus maze to study anxiety-like phenotype and to record the number of stargazing events per minute.

We have included this important information to the Methods section as well as the order of testing to ensure clarity (page 18 lines 17 – 26).

- 3- The number of animals in some behavioral experiments seem low to ensure statistical power, such as Fig. 1b, 1c and 1d

The reasons for the differences in the numbers of animals used for some of the behavioural tests are as outlined above in response to Question 2, i.e., the second smaller cohort of mice was used for additional experiments, as the first cohort of mice was used for other longitudinal experiments (rotarod and open field). Figures 1b and 1d are highly statistically significant (**, $P < 0.01$) and the adequate statistical power for these tests was achieved. Indeed, *post-hoc* power calculations using actual sample

sizes, means, standard deviations, and effect sizes from Figures 1b and 1d indicate sufficient power to detect differences at 0.859 and 0.785, respectively.

It is common in mouse behavioural research to use 5-6 mice per experimental group due to ethical considerations to reduce the number of animals used (and this was the second cohort of mice which had fewer mice than the first cohort). Figure 1c data has likely not reached significance due to a large variability in the knockout group, which could not have been predicted, therefore the *post-hoc* power of the test was indeed low. We have added a sentence to the manuscript to state that the statistical power was low in this test (page 6 lines 19 – 20).

4- In other behavioral experiments, there are huge differences in the number of animals between the control and the experimental groups

We thank the reviewer for this comment.

The difference in the number of animals between genotypes is due to the Mendelian distribution of the *Tfam* allele genotypes. Knockout mice are homozygous for loxP, i.e., *Tfam*^{loxP/loxP} which would only make up a quarter of all possible *Tfam* genotypes from a cross between compound heterozygote mice *PV*^{cre}*Tfam*^{loxP/+} x *PV*^{cre}*Tfam*^{loxP/+}, as specified in the methods section (page 18 lines 6 – 8).

Therefore, knockout mice group would be approximately only 25% of the total mouse number used in the study (e.g., the littermate controls would make up 75% of the litter), but for ethical reasons we did not want to cull any littermate control mice.

This is detailed in the Methods section and the exact percentages of mouse litters for each genotype are stated in the original and the revised manuscript.

5- In the immunohistochemistry experiments, how did authors ensure all sections have similar background. Intensity measurement in immunohistochemistry experiments is not very accurate quantitative method, I wonder if there is a better method that authors could use.

We thank the reviewer once again for this important technical question.

To directly address the question around control for background, we included a secondary antibody-only (or “no primary antibody”) control section for experiments to check for any non-specific background and used 3% Sudan Black B to quench naturally occurring brain tissue autofluorescence (as specified in the Methods section page 21 lines 11 – 12 and 20 – 21). We ensured that there was little or no background which would not interfere with the intensity measurements, which are high for a true signal. Also, there is no known reason to suggest that the level of background would be higher or lower in a certain genotype group skewing the data. All data were statistically tested to ensure the differences were statistically significant. We analysed many cells per brain region per mouse to ensure the representability of the data.

In our extensive experience in analysing datasets in mitochondrial research, single-cell information is preferred to enrich the dataset. We believe that using alternative methods to measure protein expression levels, such as western blotting on a brain homogenate, is not an accurate approach in this case due to the nature of the conditional knockout affecting only parvalbumin cells, which represent a small minority of all brain cells (less than 10%). Hence, using homogenate would lead to gross underestimation of any differences in protein expression in comparison to investigating

parvalbumin neurons directly on a single-cell level, and potentially resulting in false negative results.

We wanted to gain information on a single-cell level, hence we used advanced confocal microscopy to specifically address this question. Confocal imaging data have been published by many groups, including ours, for example, Chrysostomou et al., 2015, Lax et al. 2016, Hayhurst et al., 2019, Chen et al. 2020, Smith et al. 2022, which we referenced in this manuscript.

All staining experiments were performed in one experimental batch to reduce batch-to-batch variability and all camera and laser settings on the confocal microscope were kept the same throughout imaging of all sections, as specified in the Methods section (page 22 lines 9 – 10). Cells were selected on the cell marker signal basis to minimise bias, and not using the signal of the protein of interest being studied and which was being measured. The confocal settings were optimised to reduce any under-saturated and over-saturated pixels to ensure strong signal for downstream analyses.

REVIEWERS' COMMENTS:

Reviewer #1 (Remarks to the Author):

I thank the authors for addressing my comments and suggestions. I have no further comments and recommend the manuscript for publication.

Reviewer #2 (Remarks to the Author):

This is a well revised manuscript

Reviewer #3 (Remarks to the Author):

The authors addressed all my comments and concerns.